# A Noise Sensitivity Exponent Controls Large Statistical-to-Computational Gaps in Single- and Multi-Index Models

**Leonardo Defilippis** [1]   **Florent Krzakala** [2]   **Bruno Loureiro** [1]   **Antoine Maillard** [1][3]

## Abstract

Understanding when learning is statistically possible yet computationally hard is a central challenge in high-dimensional statistics. In this work, we investigate this question in the context of single- and multi-index models, classes of functions widely studied as benchmarks to probe the ability of machine learning methods to discover features in high-dimensional data. Our main contribution is to show that a Noise Sensitivity Exponent (NSE)—a simple quantity determined by the activation function—governs the existence and magnitude of statistical-to-computational gaps within a broad regime of these models. We first establish that, in single-index models with large additive noise, the onset of a computational bottleneck is fully characterized by the NSE. We then demonstrate that the same exponent controls a statistical-computational gap in the specialization transition of large separable multi-index models, where individual components become learnable. Taken together, our results identify the NSE as a unifying property linking noise robustness, computational hardness, and feature specialization in high-dimensional learning.

## 1. Introduction

Understanding how learning algorithms identify and exploit low-dimensional structure hidden in high-dimensional data is a central problem in modern machine learning and neural network theory. *Multi-index models* have emerged as one of the main theoretical frameworks to address this question (Dudeja & Hsu, 2018; Aubin et al., 2019; Ben Arous et al., 2021; Ba et al., 2022; Abbe et al., 2022; 2023; Dandi et al., 2024a;b; Bruna & Hsu, 2025; Troiani et al., 2025; Barbier et al., 2025). They provide a flexible yet tractable abstraction for feature learning, capturing the idea that the relevant information for prediction is concentrated in a low-dimensional subspace of the ambient space $\mathbb{R}^d$.

Formally, multi-index models assume that the response variable depends on the covariates $\boldsymbol{x} \in \mathbb{R}^d$ only through a small number of linear projections,

$$y = g(\boldsymbol{W}^\star \boldsymbol{x}), \qquad \boldsymbol{W}^\star \in \mathbb{R}^{p \times d}, \qquad (1)$$

where $p \ll d$ and the link function $g : \mathbb{R}^p \to \mathbb{R}$ may itself be stochastic, for instance through an additional dependence on independent noise variables.

This modeling assumption is both natural and powerful, and it encompasses a large fraction of the benchmark problems studied in statistical learning theory and theoretical computer science. At one extreme, it includes the *linear model* ($p = 1$ and $g(z) = z$) and its noisy variants. Allowing for a general link function $g(z)$ leads to the *single-index model* ($p = 1$), also known as the *generalized linear model*, which covers classical problems such as phase retrieval and the perceptron. For $p > 1$, the framework naturally captures *two-layer neural networks*, polynomial feature models, and structured Boolean functions such as sparse parities.

Because of this generality, multi-index models have become a focal point for a large and rapidly growing body of work aimed at understanding optimization, generalization, and algorithmic limits in high-dimensional non-convex learning problems. In particular, they serve as a canonical test-bed for studying gradient-based algorithms, feature learning, and the interplay between statistical and computational barriers. This perspective has motivated extensive research across computer science, statistical physics, optimization, and learning theory, see e.g. (Saad & Solla, 1995; 1996; Barbier et al., 2019; Veiga et al., 2022; Arnaboldi et al., 2023; Bandeira et al., 2022; Collins-Woodfin et al., 2024; Damian et al., 2023; Bietti et al., 2023; Damian et al., 2022; Arnaboldi et al., 2024; Ba et al., 2022; Cui et al., 2024; Dandi et al., 2025; Moniri et al., 2023; Berthier et al., 2025; Chen et al., 2025).

[1]Departement d'Informatique, École Normale Supérieure, PSL & CNRS. [2]Information, Learning and Physics Laboratory, École Polytechnique Fédérale de Lausanne (EPFL). [3]INRIA Paris. Correspondence to: Leonardo Defilippis <leonardo.defilippis@ens.psl.eu>.

*Proceedings of the $43^{rd}$ International Conference on Machine Learning*, Seoul, South Korea. PMLR 306, 2026. Copyright 2026 by the author(s).

In line with this recent surge of interest, notions such as the *information exponent* (IE) (Ben Arous et al., 2021) or the *generative exponent* (GE) (Damian et al., 2024), aim at classifying the sample complexity of learning in high dimension. Roughly speaking, these exponents characterize whether a model with a number $\Theta(d)$ of parameters can be learned efficiently using $O(d)$ samples, or whether a larger number of samples is computationally required.

While these notions have proved useful, they also exhibit important limitations. The information exponent, in particular, turns out to be too restrictive, as it can be easily bypassed by data reuse or by training with non-square losses (Dandi et al., 2024b). The generative exponent (GE) provides a more refined classification and identifies, for GE > 2, a class of models exhibiting a genuine statistical-to-computational gap (Damian et al., 2024; 2025). However, with the notable exception of parity-like problems, functions in this class are typically fine-tuned and unstable: small perturbations of the target function often collapse them back into easier regimes (Troiani et al., 2025; Cornacchia et al., 2025). As a result, these hardness classes appear brittle and arguably non-generic.

A more natural and robust classification instead emerges from the study of symmetries of the target function. In single-index models, the presence or absence of symmetry—most notably evenness of the link function—plays a decisive role. As discussed in (Dandi et al., 2024b; Troiani et al., 2025), if the target function has no symmetry, then typically GE = 1, and any number $\Omega(d)$ of samples is enough to achieve non-trivial and efficient learning of the target. Conversely, when a symmetry is present (e.g. even functions in the single-index case), typically GE = 2, and learnability is only possible past a phase transition in the regime of $\Theta(d)$ samples (Barbier et al., 2019; Aubin et al., 2019; Troiani et al., 2025). This distinction is both more natural and more robust, as symmetries are ubiquitous in real data and nature.

In this work, we focus first on single-index models with GE = 2, which are generic, symmetry-induced, and allow both weak and strong forms of learnability with a linear number of samples in the ambient dimension, both information-theoretically and with efficient algorithms (Barbier et al., 2019). We show that even within this apparently favorable regime, significant statistical-to-computational gaps can emerge in the presence of high noise, through a mechanism that has so far not been explored in this generality.

Further, we extend our analysis to separable multi-index models with generic activations, sometimes dubbed committee machines in the statistical physics literature – see e.g. (Schwarze & Hertz, 1993; Schwarze, 1993; Aubin et al., 2019; Barbier et al., 2025) and references therein. We show that large computational–statistical gaps for *specialization*,

i.e. the distinct learning of the different features of the target, arise for the same structural reason identified in noisy single-index models, pointing to a common mechanism governing efficient learnability across both settings.

## 2. Setting, definitions, and related works

### 2.1. Target function

Formally, multi-index models assume that the response variable depends on the covariates $\boldsymbol{x} \in \mathbb{R}^d$ only through a small number of linear projections,

$$y(\boldsymbol{x}) = g(\boldsymbol{W}^\star \boldsymbol{x}), \qquad \boldsymbol{W}^\star \in \mathbb{R}^{p \times d}, \qquad (2)$$

where $p \ll d$ and the link function $g : \mathbb{R}^p \to \mathbb{R}$ may itself be stochastic, for instance through an additional dependence on independent noise variables. In the following, we further assume that $\boldsymbol{W}^\star = (\boldsymbol{w}_1^\star, \cdots, \boldsymbol{w}_p^\star)$ has been generated from a standard Gaussian distribution $\boldsymbol{w}_k^\star \overset{\text{i.i.d.}}{\sim} \mathcal{N}(0, \mathrm{I}_d/d)$, such that $\mathbb{E}\|\boldsymbol{w}_k^\star\|_2^2 = 1$.

**High-dimensional limit —** We assume that we observe $n$ samples $\{y(\boldsymbol{x}_i), \boldsymbol{x}_i\}_{i=1}^n$, with $\boldsymbol{x}_i \overset{\text{i.i.d.}}{\sim} \mathcal{N}(0, \mathrm{I}_d)$, and we investigate the problem of reconstructing $\boldsymbol{W}^\star$ in the proportional high-dimensional limit $d, n = n(d) \to \infty$ with

$$\lim_{d \to \infty} \frac{n}{d} = \alpha > 0. \qquad (3)$$

While we focus for simplicity on isotropic Gaussian distributions for the covariates $\boldsymbol{x}_i$ and the features $\boldsymbol{w}_k^\star$, we expect that our results can be generalized to much larger classes of distributions, as we discuss in the conclusion of the paper.

**Concrete cases —** Two particular examples of the setting defined in Equation (2) will be of particular interest in the following. In all these examples, $\sigma : \mathbb{R} \to \mathbb{R}$ is a generic function, on which we will precise assumptions later on.

(i) *Single-index models* ($p = 1$) with Gaussian additive noise:

$$y(\boldsymbol{x}) = \sqrt{\lambda}\sigma(\boldsymbol{w}^\star \cdot \boldsymbol{x}) + \xi, \qquad \boldsymbol{w}^\star \in \mathbb{R}^d, \quad (4)$$

where $\xi \sim \mathcal{N}(0, 1)$ and $\lambda \geq 0$ is the *signal-to-noise* ratio.

(ii) *Separable multi-index models*, so-called *committee machines* in the statistical physics literature:

$$y(\boldsymbol{x}) = \frac{1}{\sqrt{p}} \sum_{k=1}^p \sigma(\boldsymbol{w}_k^\star \cdot \boldsymbol{x}) + \sqrt{\Delta}\xi, \qquad \boldsymbol{w}_k^\star \in \mathbb{R}^d,$$

$$(5)$$

where $\xi \sim \mathcal{N}(0, 1)$ and $\Delta > 0$ is the noise strength.

## 2.2. The Noise Sensitivity Exponent (NSE)

Crucially, the models of Equation (4), (5) depend on the single real-valued function $\sigma : \mathbb{R} \to \mathbb{R}$. We now introduce the notion of *noise-sensitivity exponent* (NSE) for such a function.

**Definition 2.1** (Noise Sensitivity Exponent). Let $\sigma : \mathbb{R} \to \mathbb{R}$ such that $|\sigma(x)| \leq C(1 + |x|^k)$ for some $C, k > 0$. The noise-sensitivity exponent is defined as

$$\beta_\star(\sigma) := \min\{\beta \in \mathbb{N} : \mathbb{E}_{z \sim \mathcal{N}(0,1)}\left[\sigma^\beta(z)(z^2 - 1)\right] \neq 0\}. \tag{6}$$

Equipped with this definition, we can state our two main results. In what follows, we informally use the term recovery to denote a data-dependent estimator achieving a non-vanishing overlap (or correlation) with the underlying signal. Formal definitions for each specific case are deferred to Definitions 3.1 and 4.1.

1. In single-index models (eq. (4)), the noise sensitivity exponent $\beta_\star$ governs how the critical sample complexity $\alpha = n/d$ for efficient recovery of $\boldsymbol{w}^\star$ scales with the signal-to-noise ratio $\lambda$. When $\beta_\star > 1$, increasing noise (i.e. decreasing $\lambda$) creates a widening gap between information-theoretic and algorithmic limits: although recovery remains statistically feasible, no known polynomial-time algorithm succeeds at the optimal scale. Specifically, information-theoretic recovery requires $\alpha_{\mathrm{WR}}^{\mathrm{IT}} = O(1/\lambda)$ samples, whereas the algorithmic threshold scales as $\alpha_{\mathrm{WR}}^{\mathrm{Alg.}} = \Theta(\lambda^{-\beta_\star})$.

2. In separable $p$-multi-index models (committee machines), the NSE controls the *specialization transition* $\alpha_{\mathrm{spec.}}$, namely the sample complexity at which individual components of the latent low-dimensional structure can be disentangled and learned. We show that, as $p \gg 1$ (*after* $d \to \infty$), $\alpha_{\mathrm{spec.}}^{\mathrm{IT}} = \Theta_p(p)$. Further, we derive that $\alpha_{\mathrm{spec.}}^{\mathrm{Alg.}} = \Theta_p(p)$ if and only if $\beta_\star = 1$. When $\beta_\star > 1$, we generically obtain that $\alpha_{\mathrm{spec.}}^{\mathrm{Alg.}} = \omega(p)$ in general, and prove that $\alpha_{\mathrm{spec.}}^{\mathrm{Alg.}} = \Theta_p(p^{\beta_\star})$ in the case of even activations. This unveils a large computational-statistical gap for specialization if $\beta_\star > 1$.

## 2.3. Examples, and related exponents

We briefly illustrate the various exponents introduced in the literature and clarify their respective roles. Consider a single-index model of the form $y = g(\boldsymbol{w}^\star \cdot \boldsymbol{x})$. The *information exponent* (IE) (Ben Arous et al., 2021) is defined as the degree of the first non-zero coefficient in the Hermite expansion of the link function $g$. More precisely, letting $\mathrm{He}_k(z)$ denote the k-th Hermite polynomial, we have

$$\mathrm{IE}(g) = \min\{k \in \mathbb{N} : \mathbb{E}_{z \sim \mathcal{N}(0,1)}[g(z)\mathrm{He}_k(z)] \neq 0\} \tag{7}$$

For example, $g(z) = \tanh(z)$ has IE $= 1$ due to its linear component, whereas $g(z) = z^2 - 1$ has IE $= 2$, corresponding to the second Hermite polynomial.

Early analyses showed that, when training with square loss and single-pass gradient-based algorithms, learning dynamics are strongly influenced by the information exponent: roughly speaking, the time required to weakly recover $\boldsymbol{w}^\star$ scales as $d^{\mathrm{IE}-1}$ (up to polylogarithmic factors). However, subsequent work demonstrated that the IE (as well as the closely related leap exponent (Abbe et al., 2022; 2023)) is not a fundamental obstruction to learning. Indeed, these limitations can be overcome by reusing data (Dandi et al., 2024b; Arnaboldi et al., 2024; Lee et al., 2024), by adding small perturbations to the target (Cornacchia et al., 2025), or by modifying the loss function (Troiani et al., 2025).

The *generative exponent* (GE) was introduced by (Damian et al., 2024) to address these shortcomings:

$$\mathrm{GE}(g) = \inf_{\mathcal{T}} \mathrm{IE}(\mathcal{T} \circ g) \tag{8}$$

Informally, it captures the effective information exponent after allowing for optimal preprocessing $\mathcal{T}(y)$ of the labels. In the single-index setting, except for carefully fine-tuned constructions (Damian et al., 2024)—which are themselves unstable under perturbations (Cornacchia et al., 2025)—the classification induced by the GE is particularly simple: non-symmetric functions typically satisfy GE $= 1$, while even functions generically satisfy GE $= 2$. In both cases, the target can be recovered efficiently with $n = O(d)$ samples.

From this perspective, the landscape of learnability in single-index models largely reduces to a symmetry-based dichotomy: non-symmetric targets are learnable without a phase transition, whereas symmetric targets exhibit a critical threshold $\alpha_c = n_c/d$ for weak recovery of $\boldsymbol{w}^\star$. This picture is consistent with classical results from the statistical physics literature on single-index and perceptron-like models (Gardner & Derrida, 1989; Schwarze & Hertz, 1992b; Zdeborová & Krzakala, 2016): what matters is whether or not one has to break a symmetry on $g$ to access $\boldsymbol{w}^\star$ (Dandi et al., 2024b).

The noise sensitivity exponent introduced in the present work, defined in Equation (6), refines this understanding by revealing a new and independent source of computational hardness. In particular, even within the class of symmetry-induced models with GE $= 2$, the NSE distinguishes between single-index targets that remain computationally accessible in the presence of large noise and those that exhibit genuine statistical-to-computational gaps. A similar gap emerges in learning the independent features of a noiseless but large-width separable multi-index target, where in this case the role played by the noise is played by the unlearned directions. This provides a finer classification of learnability that goes beyond symmetry alone and highlights the role

of noise and target width in shaping computational barriers. To make it more concrete, we can categorize the different values of the NSE $\beta_\star$ as follows.

- $\beta_\star = 1$. This condition is equivalent to IE $= 2$, and encompasses all functions with a non-vanishing $\mathrm{He}_2$-coefficient in their Hermite decomposition.

- $\beta_\star = 2$. This category captures the majority of functions with zero second Hermite coefficient. This includes higher-order Hermite polynomials $\sigma(z) = \mathrm{He}_{2k}(z)$ (with $k > 1$). In Appendix C we demonstrate that all polynomials of degree $< 20$ satisfy $\beta_\star \leq 2$.

- $\beta_\star > 2$. While existing, such functions are usually fine-tuned. In Appendix C, we construct examples with $\beta_\star \in \{3, 4\}$;

- $\beta_\star = \infty$. These functions correspond to extremely fine-tuned models with generative exponent GE $> 2$ (Damian et al., 2024), which remain unlearnable in the proportional regime $n = \Theta(d)$. We briefly outline the potential generalization to a broader setting GE $\geq 2$ at the end of Section 6.

Interestingly, the definition of the NSE is connected to the notion of *superorthogonality* employed in the analysis of Oko et al. (2024). In particular, their Proposition 13 readily implies that, for any integer $K > 0$, there exists a (polynomial) link function $\sigma$ with NSE $\beta_\star(\sigma) > K$.

## 2.4. Further related works

Training multi-index models typically leads to non-convex optimization problems. As a result, they have long served as a canonical test-bed for understanding the behavior of neural networks and gradient-based algorithms in high-dimensional, non-convex settings (Saad & Solla, 1995; 1996; Ben Arous et al., 2021; Abbe et al., 2022; 2023; Veiga et al., 2022; Ba et al., 2022; Arnaboldi et al., 2023; Collins-Woodfin et al., 2024; Damian et al., 2023; Bietti et al., 2023; Moniri et al., 2023; Berthier et al., 2025). A central question in this literature is that of *weak learnability*: how many samples are required to obtain a predictor that performs strictly better than random guessing.

Weak learnability can be studied from both a *statistical* perspective—allowing arbitrary, possibly exponential-time algorithms—and a *computational* perspective, where attention is restricted to specific algorithmic classes such as first-order or query-based methods. In the single-index case ($p = 1$), weak learnability has been extensively studied under probabilistic assumptions on the data and signal (e.g. Gaussian covariates and random weights). The information-theoretic threshold for weak recovery in the high-dimensional limit

was characterized in (Barbier et al., 2019). On the algorithmic side, sharp computational thresholds were derived for broad classes of first-order iterative algorithms, including approximate message passing and spectral methods (Mondelli & Montanari, 2019; Luo et al., 2019; Celentano et al., 2020; Maillard et al., 2020a; 2022).

A central and widely studied problem in theoretical computer science is to understand statistical–computational gaps: regimes where inference or learning is information-theoretically possible, yet conjectured to be impossible for any polynomial-time algorithm. This question arises across planted and average-case problems, and lies at the interface of complexity theory, statistics, and learning theory (see e.g. (Bandeira et al., 2018; Zdeborová & Krzakala, 2016), and the recent surveys (Gamarnik et al., 2022; Wein, 2025)). From this viewpoint, the goal is to characterize the precise boundary between what is achievable with unlimited computation and what remains feasible under algorithmic constraints, and to identify generic mechanisms responsible for computational hardness (Bandeira et al., 2022; Chen et al., 2025).

Complementary lower bounds were established under restricted computational models. In particular, (Damian et al., 2022) proved that, in the Correlational Statistical Query (CSQ) model—allowing queries of the form $\mathbb{E}[y\varphi(\boldsymbol{x})]$—learning requires $n \gtrsim O(d^{\max(1,\ell/2)})$ samples, where $\ell$ is the *information exponent* (Ben Arous et al., 2021), defined as the smallest non-zero degree in the Hermite expansion of the link function $g$. The notion of *staircase functions* introduced in (Abbe et al., 2022; 2023) refined this picture by showing that, under CSQ or online SGD, feature directions may be learned sequentially, with the sample complexity governed by the so-called *leap*, which captures the largest jump in Hermite degree conditioned on previously learned directions. Related results were obtained under the more expressive Statistical Query (SQ) model in (Damian et al., 2024), which introduced the *generative exponent* $\kappa \leq \ell$ and established lower bounds of the form $n \gtrsim O(d^{\max(1,\kappa/2)})$.

While these notions successfully explain a range of computational barriers, they are often brittle: except for notable cases such as parity functions, models with large exponents typically require fine-tuned cancellations and are unstable under perturbations of the target function. Beyond the single-index setting, rigorous results for general multi-index models ($p > 1$) remain scarce, as learnability depends sensitively on how the different directions are coupled by the link function. One notable exception is the class of *separable* multi-index models, or *committee machines*, where the target is a sum of independent single-index components. For this model, that has also been studied extensively in the statistical physics literature (Schwarze & Hertz,

1992b;a; Monasson & Zecchina, 1995; Barbier et al., 2025), both algorithmic and hardness results have been established (Aubin et al., 2019; Diakonikolas et al., 2020; Goel et al., 2020; Chen et al., 2022; Troiani et al., 2025), revealing rich statistical-to-computational trade-offs.

## 3. Statistical-computational gap in even single-index models at high noise

In this section, we assume we are given $n$ samples $\mathcal{D} := \{(\boldsymbol{x}_i, y_i)_{i \in [n]}\}$ where $y_i = y(\boldsymbol{x}_i)$ is generated by the single-index model of eq. (4) with $\sigma$ an *even* function. We consider the question of *weak recovery*, i.e. the existence of an estimator $\hat{\boldsymbol{w}}(\mathcal{D})$ that correlates with $\boldsymbol{w}^\star$ better than a random guess.

**Definition 3.1** (Weak recovery). Recall that $p = 1$ in single-index models, so $\boldsymbol{w}^\star \in \mathbb{R}^d$. Given an estimator $\hat{\boldsymbol{w}}(\mathcal{D}) \in \mathbb{R}^d$, we say that $\hat{\mathbf{w}}$ achieves *weak recovery* if there exists $\varepsilon > 0$ such that, with probability $1 - o_d(1)$:

$$\hat{\mathbf{w}}(\mathcal{D}) \neq 0 \quad \text{and} \quad \frac{|\hat{\mathbf{w}}(\mathcal{D}) \cdot \boldsymbol{w}^\star|}{\|\hat{\mathbf{w}}(\mathcal{D})\|_2 \|\boldsymbol{w}^\star\|_2} \geq \varepsilon.$$

**Computational weak recovery —** We study computational weak recovery by assessing the performance of Approximate Message Passing (AMP) algorithms (Donoho et al., 2009; Rangan, 2011). This class of algorithms algorithms are optimal among first-order iterative methods (Celentano et al., 2020; Montanari & Wu, 2024), and are widely employed to delimitate the computational limits of learnability (Zdeborová & Krzakala, 2016; Bandeira et al., 2018). The following theorem establishes that the NSE governs the fundamental computational limits of learning the single index model of eq. (4) for small values of the signal-to-noise ratio $\lambda$.

**Theorem 3.2.** *Consider the single-index model of eq. (4). Assume that the even function $\sigma$ satisfies the assumptions of Definition 2.1, and has NSE $\beta_\star < \infty$. Then, the optimal AMP algorithm achieves weak recovery (see Definition 3.1) exactly for $\alpha > \alpha_{\mathrm{WR}}^{\mathrm{Alg.}}$, where the weak recovery threshold $\alpha_{\mathrm{WR}}^{\mathrm{Alg.}}$ satisfies, as $\lambda \to 0$:*

$$\alpha_{\mathrm{WR}}^{\mathrm{Alg.}} = \Theta(\lambda^{-\beta_\star})$$

In Fig. 1 we plot the computational thresholds for weak recovery for different activations $\sigma$, as a function of $\lambda$. They are obtained by solving numerically the equations obtained for $\alpha_{\mathrm{WR}}^{\mathrm{Alg.}}$ in (Mondelli & Montanari, 2019; Maillard et al., 2020a) for generic single-index models.

**Information-theoretic weak recovery —** Finally, the following establishes that information-theoretic recovery is, on the other hand, insensitive to the NSE.

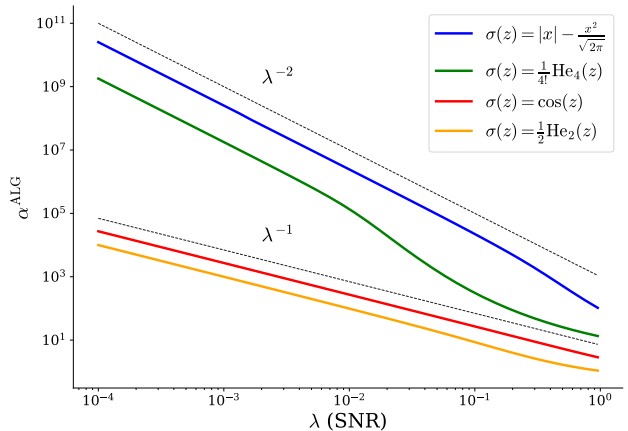

*Figure 1.* Examples of computational weak recovery thresholds $\alpha_{\mathrm{WR}}^{\mathrm{Alg.}}$ (Mondelli & Montanari, 2019; Barbier et al., 2019) as a function of the signal-to-noise ratio $\lambda$, for the single-index model of eq. (4).

**Theorem 3.3.** *Under the same setting as Theorem 3.2, there exists an estimator $\hat{\boldsymbol{w}}(\mathcal{D})$ that achives weak recovery as soon as $\alpha > \alpha_{\mathrm{WR}}^{\mathrm{IT}}$. Moreover, this information-theoretic weak recovery threshold satisfies $\alpha_{\mathrm{WR}}^{\mathrm{IT}} = \Theta(\lambda^{-1})$. Consequently, for $\beta_\star > 1$, there is a wide statistical-to-computational gap as $\lambda \to 0$:*

$$\alpha_{\mathrm{WR}}^{\mathrm{Alg.}} - \alpha_{\mathrm{WR}}^{\mathrm{IT}} = \Theta(\lambda^{-\beta_\star}).$$

## 4. The specialization transition in large separable multi-index models

We now consider the model of eq. (5), i.e. a *separable* multi-index function. Without loss of generality, we assume that $\mathbb{E}_{z \sim \mathcal{N}(0,1)}[\sigma(z)] = 0$. Again, our goal is to quantify the difficulty of learning this function from the data $\mathcal{D} := \{\boldsymbol{x}_i, y_i = y(\boldsymbol{x}_i)\}_{i=1}^n$. The computational-statistical gap we describe in this case concerns the *specialization* of solutions.

We first define the notions of weak recovery and specialization for separable multi-index models, analogous to those established in (Aubin et al., 2019).

**Definition 4.1** (Weak recovery and Specialization). For $\boldsymbol{W} \in \mathbb{R}^{p \times d}$, we denote $\Theta(\boldsymbol{W}) \in \mathbb{R}^{p \times d}$ such that $\Theta(\boldsymbol{W})_a = \boldsymbol{w}_a / \|\boldsymbol{w}_a\|_2$ if $\boldsymbol{w}_a \neq 0$, and $\Theta(\boldsymbol{W})_a = 0$ otherwise. Let $\hat{\boldsymbol{W}} = \hat{\boldsymbol{W}}(\mathcal{D}) \in \mathbb{R}^{p \times d}$ be an estimator of $\boldsymbol{W}^\star$. Then, $\hat{\boldsymbol{W}}$ is said to achieve *weak recovery* with overlap $\varepsilon$ if there exists $\varepsilon > 0$, $\boldsymbol{v} \in \mathbb{R}^p$, with $\|\boldsymbol{v}\|^2 = 1$, such that, with probability $1 - o_d(1)$,

$$\|\Theta(\hat{\boldsymbol{W}})[\Theta(\boldsymbol{W}^\star)]^\top \boldsymbol{v}\|_2 \geq \varepsilon. \tag{9}$$

If the only such vectors $\boldsymbol{v}$ satisfying this property are in $\mathrm{span}(\mathbf{1}_p)$, where $\mathbf{1}_p = (1, \ldots, 1)^\top \in \mathbb{R}^p$, the estimator

$\hat{W}$ is said to be *unspecialized*. Conversely, if there exists $v \perp \mathbf{1}_p$ satisfying eq. (9), then $\hat{W}$ is said to be *specialized*.

Informally, an *unspecialized* solution is an estimator that puts the same value for all the weights $\{w_k\}_{k=1}^p$, i.e. it fits the dataset $\mathcal{D}$ by a function

$$\hat{y}(\boldsymbol{x}) = \sqrt{p}\sigma(\hat{w} \cdot \boldsymbol{x}), \qquad (10)$$

that depends on a *single* weight vector $w$. Consequently, any solution that breaks this permutation symmetry is defined as *specialized*. Crucially, this definition of specialization is strictly weaker than that of identifiability, corresponding to mutually exclusive recovery of weights (*i.e.*, each row of the estimator correlates with one, and only one, ground-truth vector $w_k^\star$, $k \in [p]$).

Analogously to the single-index case, we define $\alpha_{\text{WR}}^{\text{IT}}$ as the information-theoretic weak recovery threshold and $\alpha_{\text{WR}}^{\text{Alg.}}$ as the threshold for computationally-efficient weak recovery (i.e. the threshold achieved by the AMP algorithm). Likewise, we define $\alpha_{\text{spec.}}^{\text{IT}}$ and $\alpha_{\text{spec.}}^{\text{Alg.}}$ as the smallest sample complexities necessary for specialization. Computational weak recovery for multi-index models in the proportional regime $n \asymp d$ (and $p = \Theta_d(1)$ as $d \to \infty$) has been studied by (Troiani et al., 2025). In Appendix D we show two distinct behaviors for the model of eq. (5), for a given $p \geq 1$:

- When $\sigma$ is even, weak recovery yields the full subspace $\text{span}(\boldsymbol{W}_\star)$ simultaneously; therefore, the specialization transition coincides with the weak recovery one ($\alpha_{\text{spec.}}^{\text{Alg.}} = \alpha_{\text{WR}}^{\text{Alg.}}$).

- When $\sigma$ is non-even, under mild assumptions, an unspecialized solution can be efficiently achieved with arbitrarily small sample complexity ($\alpha_{\text{WR}}^{\text{Alg.}} = 0$), while specialization requires $\alpha_{\text{spec.}}^{\text{Alg.}} > 0$.

Our first result concerns even functions $\sigma$: there, according to the discussion above, weak recovery and specialization are essentially equivalent. We demonstrate that this specialization transition is intrinsically governed by the NSE of the link function, via a mechanism analogous to the one observed in single-index models.

**Theorem 4.2.** *Consider the multi-index model of eq.* (5), *with $\sigma$ an **even** function with NSE $\beta_\star < \infty$ given in Definition 2.1. Then we have, as $p \to \infty$:*

$$\alpha_{\text{WR}}^{\text{Alg.}} = \alpha_{\text{spec.}}^{\text{Alg.}} = \Theta_p(p^{\beta_\star})$$

The sketch of proof of Theorem 4.2 is given in Section 5.2, with some details deferred to Appendix D.

When $\sigma$ is not symmetric, the situation is more complex, as in general weak recovery is no longer equivalent to specialization. For instance, it was predicted by (Aubin et al.,

2019) that specialization (but not weak recovery) is statistically possible but computationally hard for $\alpha \gtrsim \Theta(p)$ in a closely-related model where $\sigma(x) = \text{sign}(x)$ and one observes $\text{sign}(y(\boldsymbol{x}))$. These findings were generalized in (Citton et al., 2025) which predicts, under a so-called annealed approximation, a transition for computationally feasible specialization depending on the Hermite decomposition of the activation $\sigma$. Here we confirm these predictions without relying on such an approximation, and complement them with sharp thresholds for the information-theoretic specialization.

In the following result, which holds as $p \to \infty$, we emphasize that we consider weak recovery and specialization with an overlap $\varepsilon > 0$ that *does not go to 0* as $p \to \infty$: informally, we must recover a non-zero fraction of the target weights.

**Result 4.3.** *Consider the multi-index model of eq.* (5), *with $\sigma$ a non-linear function having NSE $\beta_\star$ per Definition 2.1. Then, as $p \to \infty$ (after $n, d \to \infty$):*

- *If $\alpha = o(p)$, the AMP estimator for $\boldsymbol{W}^\star$ achieves weak recovery (for a finite $\varepsilon > 0$ as $p \to \infty$) only possibly in the* unspecialized *direction (see Def. 4.1).*

- *If $\beta_\star = 1$, the AMP estimator achieves specialization for $\alpha > \alpha_{\text{spec.}}^{\text{Alg.}} = \Theta(p)$. On the other hand, if $\beta_\star > 1$ then the AMP iterates remain unspecialized as $p \to \infty$ for any $\alpha = o(p^{3/2})$.*

- *$\alpha_{\text{spec.}}^{\text{IT}} \leq p(1 + \varepsilon)$ for any $\varepsilon > 0$ as $p \to \infty$. Thus, for any $\beta_\star$, the statistically-optimal estimator achieves weak recovery and is specialized for $\alpha \gtrsim p$.*

*Remark* 4.4. A few remarks about this result are in order.

- Result 4.3 relies on $(i)$ a large-$p$ expansion of the information-theoretic and AMP performances using statistical-physics tools, and $(ii)$ a symmetry assumption among the $p$ hidden units of the model (sometimes dubbed *committee symmetry* in the statistical physics literature), a classical hypothesis in the study of separable multi-index models, see e.g. (Aubin et al., 2019; Barbier et al., 2025) and references therein. For these reasons, we do not state it as a theorem.

- The scaling $o(p^{3/2})$ for the non-specialization of the AMP solution is likely sub-optimal, and a byproduct of the bound on the error terms in the $p \to \infty$ expansion of AMP's performance.

- Our derivation actually suggests that for $\alpha > p(1 + \varepsilon)$, the statistically-optimal estimator is not only specialized, but achieves *perfect recovery* of $\boldsymbol{W}^\star$.

- The condition that $\sigma$ is non-linear is natural, since specialization is not present for linear activations (Aubin et al., 2019).

The derivation of Result 4.3 is sketched in Section 5.2, with some details deferred to Appendix E. Result 4.3 shows the emergence of a large statistical-to-computational gap for specialization in large-width separable multi-index models, whenever $\beta_\star > 1$. Refining this result, Theorem 4.2 precises the scale of this gap as a function of $\beta_\star$ for even activations. While Result 4.3 also establishes the existence of this gap for generic activations, it falls short of providing its scale when $\beta_\star > 1$, contrary to the even case: this naturally leads to the following question.

**Open Question 4.5.** *In the setting of Result 4.3, and as $p \gg 1$, do we have*

$$\alpha_{\text{spec.}}^{\text{Alg.}} = \Theta(p^{\beta_\star})?$$

One natural way towards Question 4.5 would be to refine the large-$p$ expansion of the AMP iterates we use to derive Result 4.3 to higher order as $p \to \infty$. This is a technically challenging endeavor, which we leave for future work.

# 5. Sketch of proofs and derivations

We sketch here the proof of Theorems 3.2 and 3.3, with more detailed derivations in Appendices A and B respectively.

## 5.1. Single-index models at large noise

**Theorem 3.2 –** The proof is based on a large noise $\lambda \to 0$ expansion of the proven formula for the weak recovery threshold in (Mondelli & Montanari, 2019; Barbier et al., 2019):

$$\left(\alpha_{\text{WR}}^{\text{Alg.}}\right)^{-1} = \mathbb{E}_y \left[ (\mathbb{E}[z^2 - 1|y])^2 \right], \tag{11}$$

with $z \sim \mathcal{N}(0,1)$ and $y \sim \mathcal{N}(\sqrt{\lambda}\sigma(z), 1)$. We consider the expansions of the involved terms around $\lambda = 0$, in order to characterize the leading order of $\alpha_{\text{WR}}^{\text{Alg.}}$ at small SNR $\lambda$. Denote with $\mathsf{Z}(y)$ the PDF of $y$:

$$\mathsf{Z}(y) := \frac{1}{\sqrt{2\pi}} \mathbb{E}_{z \sim \mathcal{N}(0,1)} \left[ \frac{\exp\left(-(y - \sqrt{\lambda}\sigma(z))^2\right)}{2} \right].$$

Leveraging Taylor's theorem, we are able to show that, as $\lambda \to 0$

$$\frac{1}{\alpha_{\text{WR}}^{\text{Alg.}} \lambda^{\beta_\star}} \to \frac{1}{\beta_\star!} \left( \mathbb{E}_z[g^{\beta_\star}(z)(z^2 - 1)] \right),$$

which yields the result.

**Theorem 3.3 –** Our argument is based on the analysis of the *free entropy* of the model, provided in (Barbier et al., 2019), and the characterization of the associated information-theoretic threshold in the large noise regime.

In this work it is proven that information-theoretic weak recovery is achieved if and only if the global maximizer $m = m(\alpha) \in [0,1]$ of the following functional $f_{\text{RS}}$ is non-zero:

$$f_{\text{RS}}(m) := m + \log(1 - m) + 2\alpha\Psi_{\text{out}}(m), \tag{12}$$

$$\Psi_{\text{out}}(m) := \mathbb{E}_{W,V,Y} \log \mathbb{E}_w \left[ \mathsf{P}(Y|\sqrt{m}V + \sqrt{1-m}w) \right],$$

with $V, W, w \sim \mathcal{N}(0,1)$ and

$$Y \sim \mathcal{N}\left( \sqrt{\lambda}\sigma(\sqrt{m}V + \sqrt{1-m}W), 1 \right).$$

We proceed with a Taylor expansion around $\lambda = 0$ of eq. (12), analogously to the proof of Theorem 3.2. We refer to Appendix B for details: we find that, up to constant terms in $m$:

$$\Psi_{\text{out}}(m) = \lambda \sum_{k \geq 0} \frac{c_k^2}{k!} m^k + \mathcal{O}(\lambda^{3/2}),$$

where $c_k := \mathbb{E}_z[\sigma(z)\text{He}_k(z)]$ and $\text{He}_k(z)$ the probabilist's Hermite polynomial of degree $k$. Finally, given suitable choices of constants $C, D > 0$, we are able to show that: (i) for $\alpha\lambda > C$ there exist $m \in (0,1]$ such that $f_{\text{RS}}(m) > f_{\text{RS}}(0)$; (ii) for $\alpha\lambda < D$, $f_{\text{RS}}(m) < f_{\text{RS}}(0)$ for all $m \in (0,1]$. This yields

$$D\lambda^{-1} \leq \alpha_{\text{WR}}^{\text{IT}} \leq C\lambda^{-1}.$$

## 5.2. Separable multi-index models

We sketch here the proofs and derivation of Theorem 4.2 and Result 4.3. Details are defered to Appendix D and E. In both cases, we assume without loss of generality that $\mathbb{E}_{z \sim \mathcal{N}(0,1)}[\sigma(z)] = 0$.

**Theorem 4.2 –** Our proof is based on a large-$p$ expansion of the formula proven for the weak recovery threshold as Lemma 4.1 in (Troiani et al., 2025). First, recall (see Appendix D) that we showed that the results of this work imply that $\alpha_{\text{WR}}^{\text{Alg.}} = \alpha_{\text{spec.}}^{\text{Alg.}}$, we therefore focus on the weak-recovery threshold. As a consequence of the permutational symmetry of the indices, we are able to show that

$$\alpha_{\text{WR}}^{\text{Alg.}} = \mathbb{E}_Y \left[ (\mathbb{E}[z_1^2 - 1|Y])^2 \right], \tag{13}$$

where $\mathbf{z} \sim \mathcal{N}(0, \text{I}_p)$ and $Y = p^{-1/2} \sum_{k=1}^p \sigma(z_k) + \sqrt{\Delta}\xi$, $\xi \sim \mathcal{N}(0,1)$. The proof is obtained adapting the arguments in Appendix A for single-index models. In particular, as a consequence of the central limit theorem, in the limit $p \to \infty$ the variable $\frac{1}{\sqrt{p}} \sum_{k=2}^p \sigma(z_k) + \sqrt{\Delta}\xi$ behaves as Gaussian noise, and we can show that the separable model effectively corresponds to a single-index problem (4) with SNR $\lambda = 1/p$.

**Result 4.3 –** We give here an informal sketch of the main ideas of the derivation, details being given in Appendix E.

We start from the results of (Aubin et al., 2019; Troiani et al., 2025): essentially, these works shows the role of a function, dubbed *replica-symmetric potential*, of a symmetric matrix $\boldsymbol{q} \in \mathbb{R}^{p \times p}$ with $0 \preceq \boldsymbol{q} \preceq \mathrm{I}_p$, and defined as follows:

$$
\begin{cases}
f(\alpha; \boldsymbol{q}) & := \frac{1}{2} \mathrm{Tr}[\boldsymbol{q}] + \frac{1}{2} \log \det[\mathrm{I}_p - \boldsymbol{q}] + \alpha \Psi(\boldsymbol{q}), \\
\Psi(\boldsymbol{q}) & := \int \mathrm{d}y\, \mathbb{E}_{\boldsymbol{\xi} \sim \mathcal{N}(0, \mathrm{I}_p)} \left[ I(y, \boldsymbol{\xi}; \boldsymbol{q}) \log I(y, \boldsymbol{\xi}; \boldsymbol{q}) \right].
\end{cases}
$$

Here $I(y, \boldsymbol{\xi}; \boldsymbol{q})$ is the PDF of $y = p^{-1/2} \sum_{k=1}^{p} \sigma(z_k)$, with $\boldsymbol{z} = (\mathrm{I}_p - \boldsymbol{q})^{1/2} \boldsymbol{Z} + \boldsymbol{q}^{1/2} \boldsymbol{\xi}$ and $\boldsymbol{Z} \sim \mathcal{N}(0, \mathrm{I}_p)$.

Two important properties of $f(\alpha; \boldsymbol{q})$ are shown in these works, which we state informally:

1. Weak recovery is possible if and only if $\boldsymbol{q} = 0$ is not the global maximum $\boldsymbol{q}^\star$ of $f(\alpha; \boldsymbol{q})$. At a finer level, $\boldsymbol{q}^\star$ is directly related to the correlation $\varepsilon$ achievable with the signal $\boldsymbol{W}^\star$, see eq. (9). Similarly, specialization is possible if $\boldsymbol{q}^\star \notin \mathrm{span}(\mathbf{1}_p \mathbf{1}_p^\top)$.

2. Computational weak recovery (i.e. through an AMP algorithm) is possible if and only if $\boldsymbol{q} = 0$ is not a *local* maximum of $f(\alpha; \boldsymbol{q})$. Computational specialization is characterized similarly by the local stability of unspecialized solutions (i.e. $\boldsymbol{q} \in \mathrm{span}(\mathbf{1}_p \mathbf{1}_p^T)$).

The derivation of Result 4.3 uses then a $p \to \infty$ expansion of this replica-symmetric potential, under a symmetry assumption on the structure of $\boldsymbol{q}$. Large-$p$ expansions of the replica-symmetric potential under such assumptions have been studied in the past, see e.g. (Schwarze & Hertz, 1992b; Monasson & Zecchina, 1995; Aubin et al., 2019; Barbier et al., 2025; Citton et al., 2025), and our derivation is close to the ones present in these works.

The leading order of this expansion yields clear predictions for the location of the global maximum $\boldsymbol{q}^\star$, as well as the stability of the unspecialized subspace $\mathrm{span}(\mathbf{1}_p \mathbf{1}_p^\top)$: these predictions are summarized in Result 4.3.

## 6. Conclusion

This work identifies a simple, activation-dependent quantity—the *noise sensitivity exponent* (NSE)—as a unifying quantifier of large statistical-to-computational gaps in two canonical high-dimensional learning benchmarks: noisy single-index models and large-width separable multi-index (committee) machines. Focusing on the generic symmetry-induced regime GE = 2, we show that noise can create substantial computational bottlenecks even when information-theoretic learning remains possible with $n = \Theta(d)$ samples.

In single-index models with additive Gaussian noise, the NSE completely governs how the algorithmic weak recovery threshold deteriorates at small signal-to-noise ratio $\lambda$: while information-theoretic weak recovery scales as $\alpha_{\mathrm{IT}} = \Theta(\lambda^{-1})$, the computational threshold scales as $\alpha_{\mathrm{Alg.}} = \Theta(\lambda^{-\beta_\star})$. Hence, whenever $\beta_\star > 1$, increasing noise opens a widening gap between what is statistically possible and what is achievable by efficient algorithms, despite the model remaining in the proportional regime.

In separable multi-index models, we connect the same exponent to the onset of *specialization*, i.e. the regime where the $p$ individual components of the planted subspace become identifiable. For even activations, weak recovery and specialization coincide and we obtain a sharp large-$p$ scaling $\alpha_{\mathrm{spec.}}^{\mathrm{Alg.}} = \alpha_{\mathrm{WR}}^{\mathrm{Alg.}} = \Theta_p(p^{\beta_\star})$, contrasting with the information-theoretic scaling $\alpha_{\mathrm{spec.}}^{\mathrm{IT}} = \Theta_p(p)$. For general activations, we predict based on analytical computations using statistical physics methods that the NSE again controls whether specialization occurs at the linear scale $\Theta(p)$ (when $\beta_\star = 1$) or is pushed to superlinear sample complexity (when $\beta_\star > 1$), yielding robust statistical-to-computational gaps beyond the even case.

**Limitations.** In terms of limitations, our results explicitly trade realism for analytic tractability, in the standard way for single- and multi-index models. First, we assume i.i.d. Gaussian covariates and random isotropic planted directions, which enable sharp asymptotics via Hermite expansions, state evolution, and replica computations. Second, our characterizations are asymptotic in a high-dimensional limit ($d, n \to \infty$ with $n/d \to \alpha$, and for multi-index models an additional large-width limit $p \to \infty$ after $d \to \infty$). While these assumptions yield clean thresholds, they do not directly quantify finite-dimensional effects, nor do they capture structured covariances, heavy-tailed features, or distributional shifts commonly present in real data. Finally, part of our multi-index analysis (Result 4.3) relies on statistical-physics arguments and a symmetry ansatz (committee symmetry), and is therefore currently non-rigorous.

We nonetheless view some of these limitations as necessary: they isolate the mechanism by which noise interacts with feature-learning structure, and allow us to identify the NSE as a sharp organizing principle for computational barriers. Moreover, we expect the Gaussian covariate assumption to be substantially relaxable. A broad line of *universality* results in high-dimensional statistics and random features suggests that, under appropriate moment and weak-dependence conditions, many threshold phenomena and asymptotic predictions derived under Gaussian designs persist far beyond the Gaussian setting (Donoho & Tanner, 2009; Bayati et al., 2015; Hu & Lu, 2022; Maillard et al., 2020b; Gerace et al., 2024; Pesce et al., 2023; Dudeja et al., 2023). We therefore anticipate that the NSE-controlled scalings identified here will remain valid for a wide class of non-Gaussian covariates, potentially after suitable preprocessing/whitening, and

leave these extensions for future work.

**Perspectives.** Several directions appear particularly natural. First, it would be important to turn Result 4.3 into a fully rigorous theorem, and to sharpen the current non-specialization range (e.g. improving the $o(p^{3/2})$ control) by developing higher-order large-$p$ expansions of state evolution. Second, our analysis suggests a concrete conjecture: for generic activations with $\beta_\star > 1$, the specialization threshold should scale as $\alpha_{\text{spec.}}^{\text{Alg.}} = \Theta(p^{\beta_\star})$ (Open Question 4.5).

The mechanisms unveiled for single-index models and committee machines suggest that the NSE governs weak recovery across a broader class of multi-index models. For instance, in separable *hierarchical* multi-index models $y = \sum_k \sqrt{\lambda_k} \sigma(\langle w_k^\star, x \rangle)$ (Ren et al., 2025; Defilippis et al., 2026), the NSE govern the computational sample complexity thresholds for the weak recovery of individual weights $w_k^\star$, at small SNR $\lambda_k$. In particular, for generic NSE $\beta_\star > 1$, in the power-law setting $\lambda_k \propto k^{-2\gamma}$, with $\gamma > 1/2$, Defilippis et al. (2026) shows that the optimal computational weak-recovery of the $k^{\text{th}}$ signal direction is achieved for $\alpha \asymp k^{2\gamma\beta_\star}$, which directly affects the scaling behavior of the mean-squared error.

Furthermore, while we focus on the generative exponent $\text{GE}(g) = 2$ case (defined in eq. 8), we note an intriguing structural parallel with the general setting $\text{GE}(g) \geq 2$, in which our definition of NSE naturally generalizes to $\beta_\star(g) = \min\{\beta \in \mathbb{N} : \mathbb{E}_z[g^\beta(z)\text{He}_{\text{GE}(g)}(z)] \neq 0\}$. Indeed, from the Statistical Query lower bound for the sample complexity threshold established in Theorem 3.2 of (Damian et al., 2024), and following reasoning analogous to the proof of our Theorem 3.2, we expect that, as $\lambda \to 0$, the critical sample complexity scales as $n \asymp \lambda^{-\beta_\star} d^{k_\star/2}$. Hence, the NSE could refine the classification of models within the same GE class. While a formal proof is left for future work, this observation suggests a potentially deeper connection between AMP-based recovery analysis and SQ bounds.

Finally, extending these mechanisms beyond Gaussian covariates and simplified architectures would help clarify how widely the NSE principle governs computational barriers in more realistic feature-learning settings. We hope that isolating the NSE as a sharp and computable driver of algorithmic hardness will facilitate such extensions and provide a clean organizing principle for statistical-to-computational gaps in high-dimensional learning.

## Acknowledgements

We would like to thank Yatin Dandi, Vittorio Erba, Lenka Zdeborova, Ilias Zadik and Theodor Misiakiewicz for insightful discussions. BL and LD were supported by the French government, managed by the National Research Agency (ANR), under the France 2030 program with the project references "ANR-23-IACL-0008" (PR[AI]RIE-PSAI) and "ANR-25-CE23-5660" (MAPLE), as well as the Choose France - CNRS AI Rising Talents program. FK acknowledge funding from the Swiss National Science Foundation grants OperaGOST (grant number 200021 200390) and DSGIANGO (grant number 225837). This work was supported by the Simons Collaboration on the Physics of Learning and Neural Computation via the Simons Foundation grant (#1257412).

## Impact Statement

This paper presents work whose goal is to advance the field of Machine Learning. There are many potential societal consequences of our work, none which we feel must be specifically highlighted here.

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

# A. Proof of Theorem 3.2

Given the optimal computational threshold (Mondelli & Montanari, 2019; Barbier et al., 2019)

$$\alpha_{\mathrm{WR}}^{\mathrm{Alg.}} := \left( \mathbb{E}[(\mathbb{E}[z^2 - 1 | Y])^2] \right)^{-1}, \tag{14}$$

for $z \sim \mathcal{N}(0, 1)$, $y \sim \mathcal{N}(\sqrt{\lambda}\sigma(z), 1)$, we aim to show that

$$\lim_{\lambda \to 0^+} \frac{\lambda^{-\beta_\star}}{\alpha_{\mathrm{WR}}^{\mathrm{Alg.}}} = \frac{\mu_{\beta_\star}^2}{\beta_\star!}, \tag{15}$$

which is a non-zero finite constant with $\mu_\beta := \mathbb{E}[\sigma^\beta(z)(z^2 - 1)]$.
For simplicity, denote the standard Gaussian density as

$$\phi(x) = \frac{e^{-x^2/2}}{\sqrt{2\pi}}. \tag{16}$$

For $y \in \mathbb{R}$, define $\mathsf{Z}(y) = \mathbb{E}_z[\phi(y - \sqrt{\lambda}\sigma(z))]$ and consider the term

$$\mathbb{E}[z^2 - 1 | y] = \frac{1}{\mathsf{Z}(y)} \mathbb{E}_z \left[ \phi(y - \sqrt{\lambda}\sigma(z))(z^2 - 1) \right]. \tag{17}$$

Taylor's theorem implies that, for $\beta \in \mathbb{N}$

$$\phi(x + \delta) = \sum_{j=1}^{\beta} \frac{\phi^{(j)}(x)}{\beta!} \delta^j + R_\delta(x) \tag{18}$$

with $\phi^{(j)}(x) = \frac{\mathrm{d}^j}{\mathrm{d}x^j}\phi(x)$ and $R_\delta(x)$ the remainder term. In some instances, it will be useful to consider the Lagrange form of the remainder $R_\delta(x) = \frac{\delta^{k+1}}{(k+1)!}\phi^{(k+1)}(x + \theta\delta)$, for some $\theta \in [0, 1]$. Recalling that, by definition of the NSE $\beta_\star$, $\mu_\beta = 0$ for $\beta < \beta_\star$,

$$\mathsf{Z}(y)\mathbb{E}[z^2 - 1 | y] = (-1)^{\beta_\star} \frac{\lambda^{\beta_\star/2}\mu_{\beta_\star}}{\beta_\star!}\phi^{(\beta_\star)}(y) + \mathbb{E}_z[(z^2 - 1)R_{\sqrt{\lambda}\sigma(z)}(y)]. \tag{19}$$

Then

$$(\alpha_{\mathrm{WR}}^{\mathrm{Alg.}})^{-1} = \int \mathrm{d}y \frac{1}{\mathsf{Z}(y)} \left( (-1)^{\beta_\star}\lambda^{\beta_\star/2}\frac{\mu_{\beta_\star}}{\beta_\star!}\phi^{(\beta_\star)}(y) + \mathbb{E}_z[(z^2 - 1)R_{\sqrt{\lambda}\sigma(z)}(y)] \right)^2. \tag{20}$$

The triangle inequality implies

$$\left| (\alpha_{\mathrm{WR}}^{\mathrm{Alg.}})^{-1} - \lambda^{\beta_\star}\frac{\mu_{\beta_\star}^2}{\beta_\star!^2} \int \mathrm{d}y \frac{1}{\mathsf{Z}(y)}(\phi^{(\beta_\star)}(y))^2 \right| \leq \int \mathrm{d}y \frac{\left( \mathbb{E}_z[(z^2 - 1)R_{\sqrt{\lambda}\sigma(z)}(y)] \right)^2}{\mathsf{Z}(y)} \tag{21}$$

$$\leq \int \mathrm{d}y \frac{\mathbb{E}_z[(z^2 - 1)^2 R_{\sqrt{\lambda}\sigma(z)}^2(y)]}{\mathsf{Z}(y)} \quad \text{(by Jensen's ineq.)} \tag{22}$$

From the left-hand side, we characterize $\lim_{\lambda \to 0^+} \int \mathrm{d}y \frac{1}{\mathsf{Z}(y)}(\phi^{(\beta_\star)}(y))^2$. For $M > 0$, define the indicator function $\mathbf{1}_M(z) := 1$ if $|\sigma(z)| < M$ and $\mathbf{1}_M(z) := 0$ otherwise. It is straightforward to show that there exists a constant $M$ such that

$\Pr(|\sigma(z)| < M) > 0$. For such choice of $M$, for all $\lambda < 1/M^2$

$$\mathbb{E}_z[\phi(y - \sqrt{\lambda}\sigma(z))] = \mathbb{E}_z[\mathbf{1}_M(z)\phi(y - \sqrt{\lambda}\sigma(z))] + \underbrace{\mathbb{E}_z[(1 - \mathbf{1}_M(z))\phi(y - \sqrt{\lambda}\sigma(z))]}_{\geq 0} \tag{23}$$

$$\geq \mathbb{E}_z[\mathbf{1}_M(z)\phi(y - \sqrt{\lambda}\sigma(z))] \tag{24}$$

$$= \phi(y)\mathbb{E}_z\left[\mathbf{1}_M(z)\exp\left(\underbrace{-\lambda\sigma^2(z)/2}_{\geq -1/2} + \underbrace{\sqrt{\lambda}\sigma(z)y}_{\geq -|y|}\right)\right] \tag{25}$$

$$\geq \phi(y)e^{-1/2 - |y|}\mathbb{E}_z[\mathbf{1}_M(z)] \tag{26}$$

$$= \phi(y)e^{-|y|}\underbrace{\frac{\Pr(|\sigma(z)| < M)}{\sqrt{e}}}_{=:C_M}. \tag{27}$$

Hence, given the integrable function $G(y) := C_M^{-1}e^{|y|}\phi(y)\mathrm{He}_{\beta_\star}^2(y)$, we have that, for sufficiently small $\lambda$,

$$\frac{1}{\mathsf{Z}(y)}(\phi^{(\beta_\star)}(y))^2 \leq \frac{e^{|y|}(\phi^{(\beta_\star)}(y))^2}{C_M\phi(y)} \leq C_M^{-1}e^{|y|}\phi(y)\mathrm{He}_{\beta_\star}^2(y) = G(y), \tag{28}$$

where $\mathrm{He}_\beta(x)$ are the probabilist's Hermite polynomials defined as

$$\mathrm{He}_\beta(x) = (-1)^\beta \phi(x)^{-1}\phi^{(\beta)}(x), \tag{29}$$

satisfying

$$\mathbb{E}_{x \sim \mathcal{N}(0,1)}[\mathrm{He}_\beta(x)\mathrm{He}_{\beta'}(x)] = \delta_{\beta\beta'}\beta!. \tag{30}$$

The right-hand side of eq. (28) is an integrable function. Therefore, by the dominated convergence theorem,

$$\lim_{\lambda \to 0^+} \int dy \frac{(\phi^{(\beta_\star)}(y))^2}{\mathsf{Z}(y)} = \int dy \lim_{\lambda \to 0^+} \frac{(\phi^{(\beta_\star)}(y))^2}{\mathsf{Z}(y)} \tag{31}$$

$$= \int dy \lim_{\lambda \to 0^+} \frac{(\phi^{(\beta_\star)}(y))^2}{\mathbb{E}_z[\phi(y - \sqrt{\lambda}\sigma(z))]} \tag{32}$$

$$= \int dy \frac{(\phi^{\beta_\star}(y))^2}{\phi(y)} \tag{33}$$

$$= \int dx \phi(x)\mathrm{He}_\beta^2(x) = \beta_\star!. \tag{34}$$

In order to complete the proof of eq. (15), we have to show that the remainder contribution in eq. (22) is $o(\lambda^{-\beta_\star})$. Define $\mathbf{1}_{\mathrm{in}}(z) = \mathbf{1}_{\lambda^{-1/4}}(z)$, and $\mathbf{1}_{\mathrm{out}}(z) = 1 - \mathbf{1}_{\mathrm{in}}$:

$$\frac{\mathbb{E}_z[(z^2 - 1)^2 R_{\sqrt{\lambda}\sigma(z)}^2(y)]}{\mathsf{Z}(y)} \tag{35}$$

$$= \frac{\mathbb{E}_z[(z^2 - 1)^2 R_{\sqrt{\lambda}\sigma(z)}^2(y)\mathbf{1}_{\mathrm{in}}(z)]}{\mathsf{Z}(y)} + \frac{\mathbb{E}_z[(z^2 - 1)^2 R_{\sqrt{\lambda}\sigma(z)}^2(y)\mathbf{1}_{\mathrm{out}}(z)]}{\mathsf{Z}(y)}. \tag{36}$$

For the first term, we substitute the Lagrange form for the remainder, which yields

$$\mathbb{E}_z \frac{(z^2 - 1)^2 R_{\sqrt{\lambda}\sigma(z)}^2(y)}{\mathsf{Z}(y)}\mathbf{1}_{\mathrm{in}}(z)$$

$$= \frac{\lambda}{(\beta_\star + 1)!^2}\mathbb{E}_z \frac{\left((z^2 - 1)(-\sigma(z))^{\beta_\star+1}\phi^{(\beta_\star+1)}(y - \sqrt{\lambda}\theta\sigma(z))\right)^2}{\mathsf{Z}(y)}\mathbf{1}_{\mathrm{in}}(z)$$

$$\leq \frac{\lambda}{(\beta_\star + 1)!^2}\mathbb{E}_z \frac{\left((z^2 - 1)(-\sigma(z))^{\beta_\star+1}\phi(y - \sqrt{\lambda}\theta\sigma(z))\mathrm{He}_{\beta_\star+1}(y - \sqrt{\lambda}\theta\sigma(z))\right)^2}{C_M\phi(y)e^{-|y|}}\mathbf{1}_{\mathrm{in}}(z) \qquad \text{by eq. (23)}$$

Using the inequality $(y + \delta)^2 \geq 3y^2/4 - 3\delta^2$ in $\phi(y - \sqrt{\lambda}\theta\sigma(z)))^2$, and the fact that there exists a constant $D$ such that $|\mathrm{He}_\beta(x + y)| \leq D(1 + |x|^\beta + |y|^\beta)$, we obtain

$$\leq \frac{\lambda}{(\beta_\star + 1)!^2}C_M^{-1}D^2 e^{-y^2/4+|y|}\mathbb{E}_z\left[(z^2 - 1)^2\sigma(z)^{2\beta_\star+2}\underbrace{e^{3\lambda\theta^2\sigma^2(z)}}_{\leq e^{3\sqrt{\lambda}}}(1 + |y|^{2\beta_\star+2} + \underbrace{(\sqrt{\lambda}\theta|\sigma(z)|)^{2\beta_\star+2}}_{\leq \lambda^{(\beta_\star+1)/2}})\mathbf{1}_{\mathrm{in}}(z)\right] \quad (37)$$

$$\overset{(\lambda \leq 1)}{\leq} C\lambda\, e^{-y^2/4+|y|}\mathbb{E}_z[(z^2 - 1)^2\sigma^{2\beta_\star+2}(z)], \tag{38}$$

for some constant $C$. The latter is an integrable function with respect to $y$, and

$$\lambda^{-\beta_\star}\int \mathrm{d}y\, \frac{\mathbb{E}_z[(z^2 - 1)^2 R^2_{\sqrt{\lambda}\sigma(z)}(y)\mathbf{1}_{\mathrm{in}}(z)]}{\mathsf{Z}(y)} = O(\lambda). \tag{39}$$

For the second contribution, we have, from the definition of the remainder in eq. (18) and using the triangle inequality,

$$\mathbb{E}_z\frac{(z^2 - 1)^2 R^2_{\sqrt{\lambda}\sigma(z)}(y)}{\mathsf{Z}(y)}\mathbf{1}_{\mathrm{out}}(z) \leq \mathbb{E}_z\frac{(z^2 - 1)^2\phi^2(y - \sqrt{\lambda}\sigma(z))}{\mathsf{Z}(y)}\mathbf{1}_{\mathrm{out}}(z) \tag{40}$$

$$+ \mathbb{E}_z\frac{(z^2 - 1)^2\left(\sum_{\beta=0}^{\beta_\star}\frac{\lambda^{\beta/2}}{\beta!}(-\sigma(z))^\beta\phi^{(\beta)}(y)\right)^2}{\mathsf{Z}(y)}\mathbf{1}_{\mathrm{out}}(z) \tag{41}$$

Looking at the right-hand side, the second term can be bounded as follows

$$\mathbb{E}_z\frac{(z^2 - 1)^2\left(\sum_{\beta=0}^{\beta_\star}\frac{\lambda^{\beta/2}}{\beta!}(-\sigma(z))^\beta(\phi^\beta(y))^2\right)^2}{\mathsf{Z}(y)}\mathbf{1}_{\mathrm{out}}(z)$$

$$= \frac{e^{-y^2}}{\mathsf{Z}(y)}\sum_{\beta,\beta'=0}^{\beta_\star}\frac{\lambda^{(\beta+\beta')/2}}{\beta!\beta'!}\mathrm{He}_\beta(y)\mathrm{He}_{\beta'}(y)\mathbb{E}_z[(z^2 - 1)^2\sigma^{2\beta}(z)\mathbf{1}_{\mathrm{out}}(z)]$$

$$\leq \frac{e^{-y^2/2+|y|}}{C_M}\sum_{\beta,\beta'=1}^{\beta_\star}\frac{\lambda^{(\beta+\beta')/2}}{\beta!\beta'!}\mathrm{He}_\beta(y)\mathrm{He}_{\beta'}(y)\mathbb{E}_z\left[(z^2 - 1)^2\sigma^{2\beta}(z)\mathbf{1}_{\mathrm{out}}(z)\right] \quad \text{(by eq. (23))}$$

$$\leq \frac{e^{-y^2/2+|y|}}{C_M}\sum_{\beta,\beta'=1}^{\beta_\star}\frac{\lambda^{(\beta+\beta')/2}}{\beta!\beta'!}\mathrm{He}_\beta(y)\mathrm{He}_{\beta'}(y)\underbrace{\mathbb{E}_z\left[(z^2 - 1)^4\sigma^{4\beta}(z)\right]^{1/2}}_{<\infty}(\mathrm{Pr}(|\sigma(z)| \geq \lambda^{-1/4}))^{1/2} \quad \text{(by Cauchy-Schwarz ineq.)}$$

which is integrable with respect to $y$ and, by Markov's inequality, the last factor $\mathrm{Pr}(|\sigma(z)| \geq \lambda^{-1/4}) \leq \mathbb{E}[|\sigma(z)|^c]\lambda^{c/4}$, for any $c \in \mathbb{N}$. The first term in the right-hand side of (40) can be treated in the following way

$$\frac{\mathbb{E}_z[(z^2 - 1)^2\phi^2(y - \sqrt{\lambda}\sigma(z))\mathbf{1}_{\mathrm{out}}(z)]}{\mathsf{Z}(y)}$$

$$= \frac{\mathbb{E}_z[(z^2 - 1)^2\phi^2(y - \sqrt{\lambda}\sigma(z))\mathbf{1}_{\mathrm{out}}(z)]}{\mathbb{E}_z[\phi(y - \sqrt{\lambda}\sigma(z))]}$$

$$\leq \frac{\left(\mathbb{E}_z[(z^2 - 1)^4\phi^3(y - \sqrt{\lambda}\sigma(z))\mathbf{1}_{\mathrm{out}}(z)]\mathbb{E}_z[\phi(y - \sqrt{\lambda}\sigma(z))]\right)^{1/2}}{\mathbb{E}_z[\phi(y - \sqrt{\lambda}\sigma(z))]} \quad \text{(by Cauchy-Schwarz ineq.)}$$

$$= \mathbb{E}_z[(z^2 - 1)^4\phi^3(y - \sqrt{\lambda}\sigma(z))\mathbf{1}_{\mathrm{out}}(z)].$$

Integrating with respect to $y$:

$$\int dy \mathbb{E}_z[(z^2-1)^4 \phi^3(y-\sqrt{\lambda}\sigma(z))\mathbf{1}_{\text{out}}(z)]$$

$$= \mathbb{E}_z[(z^2-1)^4 \mathbf{1}_{\text{out}}(z)] \int dy \phi^3(y)$$

$$\leq \left(\mathbb{E}_z[(z^2-1)^8]\Pr(|\sigma(z)| > \lambda^{-1/4})\right)^{1/2} \int dy \phi^3(y) \qquad \text{(by Cauchy-Schwartz ineq.)},$$

which is finite and $o(\lambda^{\beta/2})$ (by Markov's inequality). Having assembled all results, for $\lambda$ sufficiently small

$$\lambda^{-\beta_\star}(\alpha_{\text{WR}}^{\text{Alg.}})^{-1} = \frac{\mu_{\beta_\star}^2}{\beta_\star!} + o_\lambda(1), \tag{42}$$

which leads to the result in eq. (15) and completes the proof.

## B. Proof of Theorem 3.3

In (Barbier et al., 2019), it is proven that the information-theoretic sample complexity threshold corresponds to the smallest $\alpha$ such that there exists a non-zero maximizer $m$ to the following

$$\sup_{m \in [0,1]} \{m + \log(1-m) + 2\alpha\Psi_{\text{out}}(m)\}, \tag{43}$$

$$\Psi_{\text{out}}(m) := \mathbb{E}_{W,V,Y} \log \mathbb{E}_{w \sim \mathcal{N}(0,1)}\left[P(Y|\sqrt{m}V + \sqrt{1-m}w)\right], \tag{44}$$

with $V, W \sim \mathcal{N}(0,1)$, $Y \sim P(\cdot|\sqrt{m}V + \sqrt{1-m}W)$. For a fixed $m \in [0,1]$), consider the function

$$F(\kappa) := \mathbb{E}_{W,V,Y} \log \mathbb{E}_{w \sim \mathcal{N}(0,1)}\left[\exp\left(y - \kappa\sigma(\sqrt{m}V + \sqrt{1-m}w)\right)\right], \tag{45}$$

with $V, W \sim \mathcal{N}(0,1)$, $Y \sim \mathcal{N}(\kappa\sigma(\sqrt{m}V + \sqrt{1-m}W), 1)$. It is straightforward to show, by a change of variable $Y \to -Y$, that $F$ is even. Note that, up to constant terms in $m$, $F(\kappa) = \Psi_{\text{out}}(m; \kappa^2)$.

Furthermore, as a consequence of Taylor's theorem, there exists constant $\hat{\kappa} \in [0, \kappa]$, such that $F(\kappa) = F(0) + \frac{1}{2}F''(0)\kappa^2 + \frac{1}{24}F^{(4)}(\hat{\kappa})\kappa(\kappa - \hat{\kappa})^3$. By Lemma C.5 in Defilippis et al. (2026), there exists a finite $C > 0$, constant in $\kappa$ and $m$, such that

$$\left|\frac{1}{24}F^{(4)}(\hat{\kappa})\kappa(\kappa - \hat{\kappa})^3\right| < C\kappa^4. \tag{46}$$

Define the auxiliary function

$$Z(\kappa, v, u, y) := \frac{1}{\sqrt{2\pi}}\mathbb{E}_w\left[\exp\left(-(y - \kappa(\sigma(\sqrt{m}v + \sqrt{1-m}w) - \sigma(\sqrt{m}v + \sqrt{1-m}u)))^2/2\right)\right]. \tag{47}$$

In particular, $Z(0, v, u, y) = (2\pi)^{-1/2}e^{-y^2/2}$ and, applying the definition of the probabilist's Hermite polynomials, for all $j \in \mathbb{N}$, using the shorthand $g_u = \sigma(\sqrt{m}v + \sqrt{1-m}u)$

$$\frac{\partial^j}{\partial\kappa^j}Z(\kappa, v, u, y) = \frac{1}{\sqrt{2\pi}}\mathbb{E}_w\left[e^{-(y-\kappa(g_w-g_u))^2/2}\text{He}_j(y - \kappa(g_w - g_u))\frac{(g_w - g_u)^j}{j!}\right] \tag{48}$$

we have, after a change of variable $Y \to Y + \kappa g\sqrt{m}V + \sqrt{1-m}W)$, that $F(\kappa) = \mathbb{E}_{V,W,Y}[\log Z(\kappa, V, W, Y)]$ for $Y \sim \mathcal{N}(0,1)$ and

$$F''(0) = \mathbb{E}_{V,W,Y}\left[\frac{Z''(0,V,W,Y)}{Z(0,V,W,Y)} - \left(\frac{Z'(0,V,W,Y)}{Z(0,V,W,Y)}\right)^2\right] \tag{49}$$

$$= \mathbb{E}_Y[Y](\mathbb{E}_{V,W}[g_w] - \mathbb{E}_{V,w}[g_w]) - \frac{1}{2}\mathbb{E}_Y[(Y^2-1)]\left(\mathbb{E}_{V,W}[g_w^2] + \mathbb{E}_{V,W}[g_w^2] - 2\mathbb{E}_{V,W,w}[g_W g_w]\right) \tag{50}$$

$$= 0 - \mathbb{E}_{z \sim \mathcal{N}(0,1)}[g^2(z)] + \mathbb{E}_{(z_1, z_2) \sim \mathcal{N}(\mathbf{0}_2, C)}[\sigma(z_1)\sigma(z_2)] \tag{51}$$

with

$$C = \begin{pmatrix} 1 & m \\ m & 1 \end{pmatrix}. \tag{52}$$

In the above we used

$$\mathbb{E}_{V,W}[g_W^2] = \mathbb{E}_{W,V}[g^2(\sqrt{m}V + \sqrt{1-m}W)] = \mathbb{E}_z[g^2(z)], \tag{53}$$

$$\mathbb{E}_{V,w,W}[g_w g_W] = \mathbb{E}_V[\mathbb{E}_W[\sigma(\sqrt{m}V + \sqrt{1-m}W)]\mathbb{E}_w[\sigma(\sqrt{m}V + \sqrt{1-m}w)]] \tag{54}$$

$$= \mathbb{E}_{(z_1,z_2)\sim\mathcal{N}(\mathbf{0}_2,C)}[\sigma(z_1)\sigma(z_2)]. \tag{55}$$

Therefore, up to constant terms in $m$,[1]

$$\Psi_{\text{out}}(m;\lambda) = F(\sqrt{\lambda}) = \text{const} + \frac{\lambda}{2}\mathbb{E}_{(z_1,z_2)\sim\mathcal{N}(\mathbf{0}_2,C)}[\sigma(z_1)\sigma(z_2)] + O(\lambda^2). \tag{56}$$

By assumption, $\sigma$ is polynomially bounded and can be decomposed in the Hermite basis as

$$\sigma(z) = \sum_{k\geq 0} \frac{c_k}{k!}\text{He}_k(z), \qquad c_k := \frac{1}{k!}\mathbb{E}_{z\sim\mathcal{N}(0,1)}\left[\sigma(z)\text{He}_k(z)\right]. \tag{57}$$

Leveraging Proposition 11.31 in (O'Donnell, 2014),

$$\mathbb{E}_{(z_1,z_2)\sim\mathcal{N}(\mathbf{0}_2,C)}[\sigma(z_1)\sigma(z_2)] = \sum_{k\geq 0} \frac{c_k^2}{k!}m^k \in [0, \mathbb{E}_z[g^2(z)]], \tag{58}$$

which is a non-decreasing function of $m \in [0,1]$. Note that, since $\sigma$ has generative exponent 2, and $\mathbb{E}_z[\sigma(z)] = 0$, we have that $c_0 = c_1 = 0$ necessarily. Hence, we are interested in maximizing the quantity[2]

$$f_{\text{RS}}(m) := m + \log(1-m) + \alpha\lambda \sum_{k\geq 2} \frac{c_k^2}{k!}m^k + O(\alpha\lambda^2) \tag{59}$$

$$= \frac{1}{2}\left(\alpha\lambda c_2^2 - 1\right)m^2 + \sum_{k>2}\left(\frac{\alpha\lambda}{k!}c_k^2 - \frac{1}{k}\right)m^k + O(\alpha\lambda^2), \tag{60}$$

where we have expanded $\log(1-m)$. We show that there exist constants $C, D > 0$, such that $D\lambda^{-1} \leq \alpha_{\text{IT}} \leq C\lambda^{-1}$. If $c_2 \neq 0$

$$f_{\text{RS}}(m) = \frac{1}{2}\left(\alpha\lambda c_2^2 - 1\right)m^2 + \sum_{k\geq 4}\left(\frac{\alpha\lambda}{k!}c_k^2 - \frac{1}{k}\right)m^\ell \tag{61}$$

For $\alpha > \lambda^{-1}c_2^{-2}$, $m = 0$ becomes a minimum. In the case $c_2 = 0$, $m = 0$ is always a maximum, and $f(0) = 0$. Define, for some $\hat{m} \in (0,1)$,

$$C := -\frac{\log(1-\hat{m})}{\sum_k \frac{c_k^2}{k!}\hat{m}^k}. \tag{62}$$

Then, for $\alpha \geq C\lambda^{-2}$, $f(\hat{m}) > f(0)$ which implies that $m = 0$ is not the global maximum. Denote

$$D := \inf_{m\in(0,1]} \frac{-m - \log(1-m)}{\sum_k \frac{c_k^2}{k!}m^k} > 0, \tag{63}$$

Note that

$$\lim_{m\to 0^+} \frac{-m - \log(1-m)}{\sum_k \frac{c_k^2}{k!}m^k} = c_2^{-2} \implies \frac{1}{2\mathbb{E}_z[\sigma^2(z)]} \leq D \leq c_2^{-2} = \frac{1}{2\mathbb{E}_z[\sigma''(z)]}. \tag{64}$$

Then, for all $\alpha < D\lambda^{-1}$, $f(m \neq 0) < f(0) = 0$, *i.e.* $m = 0$ is the global maximizer.

---

[1]Recall that, by Lemma C.5, the correction $O(\lambda^2)$ is uniform in $m \in [0,1]$.

[2]Note that we are neglecting constant terms with respect to $m$.

## C. Examples

In this section we present some examples of functions corresponding to NSE $\beta_\star > 1$. Recall that Proposition 13 in Oko et al. (2024) guarantees that, for any integer $K > 0$, one can construct a function with NSE $\beta_\star > 0$. We focus on the case of even functions $\sigma : \mathbb{R} \to \mathbb{R}$. By the assumpion in Definition 2.1, $\sigma$ admits a decomposition in Hermite basis:

$$\sigma(z) = \sum_{k \geq 1} \frac{\sigma_{2k}}{(2k)!} \mathrm{He}_{2k}(z). \tag{65}$$

The case $\beta_\star = 1$ trivially corresponds to $\sigma_2 \neq 0$. Consider instead the case $\sigma_2 = 0$ and the quantity

$$\mathbb{E}[(z^2 - 1)\sigma^2(z)] \propto \frac{1}{2}\mathbb{E}[\mathrm{He}_2(z)\sigma^2(z)] = \sum_{k,h \geq 1} \frac{\sigma_{2k}\sigma_{2h}}{2(2k)!(2h)!}\mathbb{E}[\mathrm{He}_2(x)\mathrm{He}_{2k}(z)\mathrm{He}_{2h}(x)] = \langle \boldsymbol{\sigma}, \boldsymbol{H}\boldsymbol{\sigma} \rangle \tag{66}$$

with $[\boldsymbol{\sigma}]_k := \sigma_{2k}/(2k!)$ and $H_{kh} := \mathbb{E}[\mathrm{He}_2(x)\mathrm{He}_{2k}(z)\mathrm{He}_{2h}(z)]$. By definition, if $\langle \boldsymbol{\sigma}, \boldsymbol{H}\boldsymbol{\sigma} \rangle \neq 0$, the NSE $\beta_\star = 2$, otherwise $\beta_\star > 2$. Using the generating function of Hermite polynomials

$$e^{zt - t^2/2} = \sum_{k \geq 0} \frac{t^k}{k!} \mathrm{He}_k(z), \tag{67}$$

we obtain

$$\mathbb{E}[e^{z(t_1+t_2+t_3)-(t_1^2+t_2^2+t_3^2)/2}] = \sum_{k,h,j \geq 0} \frac{t_1^k t_2^h t_3^j}{k!h!j!}\mathbb{E}[\mathrm{He}_k(z)\mathrm{He}_h(z)\mathrm{He}_j(z)]. \tag{68}$$

At the same time, computing the Gaussian integral,

$$\mathbb{E}[e^{z(t_1+t_2+t_3)-(t_1^2+t_2^2+t_3^2)/2}] = e^{t_1 t_2 + t_1 t_3 + t_2 t_3} = \sum_{a,b,c \geq 0} \frac{t_1^{a+b} t_2^{a+c} t_3^{b+c}}{a!b!c!}. \tag{69}$$

By matching the terms in the two expressions we find that, defining $s = (k + h + j)/2$

$$\frac{\mathbb{E}[\mathrm{He}_k(z)\mathrm{He}_h(z)\mathrm{He}_j(z)]}{k!h!j!} = ((s-k)!(s-h)!(s-j)!)^{-1}, \tag{70}$$

if $k + h > j$ and $|k - h| < j$. In particular

$$H_{kh} = 2(2k)!(2h)!\,((k-h+1)!\,(h-k+1)!\,(h+k-1)!)^{-1}, \quad \text{if } |k - h| < 1 \tag{71}$$

$$= \begin{cases} 4k(2k)!, & k = h \\ (2\max(k,h))!, & |k - h| = 1 \\ 0 & \text{otherwise} \end{cases} \tag{72}$$

We seek to find finite degree $2m$ even polynomials with NSE $\beta_\star > 2$. For this purpose denote with $\boldsymbol{H}^{(m)} \in \mathbb{R}^{m \times m}$ the matrix generated from $\boldsymbol{H}$ as $\boldsymbol{H}^{(m)} := (H_{kh})_{k,h \in [m]})$. It is easy to find vectors $\boldsymbol{\sigma} \in \mathbb{R}^m$ such that $\langle \boldsymbol{\sigma}, \boldsymbol{H}\boldsymbol{\sigma} \rangle > 0$, for instance any higher-order even Hermite polynomial $\mathrm{He}_{2k}$ (which correspond to $[\boldsymbol{\sigma}]_h = \delta_{hk}/(2k)!$). Denote as $\boldsymbol{\sigma}_+$ a vector of this type. If we can find $\boldsymbol{\sigma}_- \in \mathbb{R}^m$ s.t. $\langle \boldsymbol{\sigma}, \boldsymbol{H}\boldsymbol{\sigma} \rangle < 0$, it is possible to construct a function with NSE larger than 2 as a fine-tuned linear combination of $\boldsymbol{\sigma}_+$ and $\boldsymbol{\sigma}_-$. Indeed, denote $a = \langle \boldsymbol{\sigma}_+, \boldsymbol{H}^{(m)}\boldsymbol{\sigma}_+ \rangle > 0$, $b = \langle \boldsymbol{\sigma}_-, \boldsymbol{H}^{(m)}\boldsymbol{\sigma}_- \rangle < 0$ and $c = \langle \boldsymbol{\sigma}_-, \boldsymbol{H}^{(m)}\boldsymbol{\sigma}_+ \rangle$

$$\langle (t\boldsymbol{\sigma}_+ + \boldsymbol{\sigma}_-), \boldsymbol{H}^{(m)}(t\boldsymbol{\sigma}_+ + \boldsymbol{\sigma}_-) \rangle = at^2 + b + 2tc \overset{!}{=} 0, \tag{73}$$

which is always solvable since $c^2 - ab > 0$. Therefore, the existence of a function with $\beta_\star > 2$ is equivalent to $\boldsymbol{H}^{(m)}$ having at least one negative eigenvalue.

Numerical verification shows that $\boldsymbol{H}^{(m)}$ is positive-definite for $m < 10$. The first negative eigenvalue appears at $m = 10$ (degree 20), with value $\approx -4.18$. This implies that all even polynomials of degree less than 20 have $\beta_\star \leq 2$.
In a similar fashion, in order to construct functions with $\beta_\star > 3$, consider two functions $\sigma_+$, $\sigma_-$ with $\beta_\star = 3$ (which can be constructed with the method just described), such that $\mathbb{E}[\sigma_+^3(z)(z^2 - 1)] > 0$ and $\mathbb{E}[\sigma_-^3(z)(z^2 - 1)] < 0$.[3] Then there exists $t \in [0, 1]$ such that $\mathbb{E}[(t\sigma_+(z) + (1-t)\sigma_-(z))^3(z^2 - 1)] = 0$, due to continuity.

---

[3]It is possible to satisfy such condition, since $\mathbb{E}[(-\sigma(z))^3(z^2 - 1)] = -\mathbb{E}[\sigma^3(z)(z^2 - 1)]$.

# D. Proof of Theorem 4.2

Given $\boldsymbol{z} \sim \mathcal{N}(\boldsymbol{0}_p, \boldsymbol{I}_p)$, $\xi \sim \mathcal{N}(0,1)$, $Y = p^{-1/2} \sum_{k=1}^p \sigma(z_i) + \sqrt{\Delta}\xi$, consider the vector $\mathbb{E}[\boldsymbol{z}|Y]$. Given $\sigma$ even, the conditional distribution $\mathsf{P}(\cdot|\boldsymbol{z})$ is even with respect to each $z_k$, $k \in [p]$, and consequently

$$\mathbb{E}[z_k|Y = y] \propto \mathbb{E}_{\boldsymbol{z}}[z_k \mathsf{P}(y|\boldsymbol{z})] = 0 \quad \forall k \in [p]. \tag{74}$$

In Theorem 3.2 of (Troiani et al., 2025) it is established that, when $\mathbb{E}[\boldsymbol{z}|Y] = 0$ a.s., the trivial subspace of $\mathrm{span}(\boldsymbol{W}^\star)$ is empty, *i.e.* there does not exist a subspace that can be efficiently learned at any sample complexity $\alpha$ (see Definition 3 in (Troiani et al., 2025)). Instead, there exists a strictly positive threshold $\alpha_{\mathrm{WR}}^{\mathrm{Alg.}} > 0$ for weak recovery, that is given by (Lemma 4.1 in (Troiani et al., 2025))

$$\alpha_{\mathrm{WR}}^{\mathrm{Alg.}} = \left( \sup_{\boldsymbol{M} \in \mathbb{S}_{m_\star}^+, \|\boldsymbol{M}\|_F = 1} \|\mathbb{E}_{Y \sim \mathcal{Z}} \boldsymbol{G}(Y) \boldsymbol{M} \boldsymbol{G}(Y)\|_F \right)^{-1}, \tag{75}$$

with $\boldsymbol{G}(y) := \mathbb{E}[\boldsymbol{z}\boldsymbol{z}^\top - \boldsymbol{I}_{m_\star} \mid Y = y]$. Exploiting again the parity of $\sigma$, it is straightforward to show that $\boldsymbol{G}(y)$ is diagonal. In fact,

$$\mathbb{E}[z_k z_h|y] \propto \int e^{-\|\boldsymbol{z}\|^2/2} \exp\left( -\left( y - \sum_{k=1}^m a_k g_k(z_k) \right)^2 /(2\Delta) \right) z_h z_k \, \mathrm{d}\boldsymbol{z} = 0. \tag{76}$$

Moreover, due to the permutational symmetry of the committee model,

$$\mathbb{E}[z_k^2 - 1|y] = \mathbb{E}[z_h^2 - 1|y], \quad \forall k, h \in [p]. \tag{77}$$

Therefore $\boldsymbol{G}(y) = \mathbb{E}[z_1^2 - 1|Y = y]\boldsymbol{I}_p$, and

$$\sup_{\boldsymbol{M} \in \mathbb{S}_{m_\star}^+, \|\boldsymbol{M}\|_F = 1} \|\mathbb{E}_{Y \sim \mathsf{Z}} \boldsymbol{G}(Y) \boldsymbol{M} \boldsymbol{G}(Y)\|_F = \mathbb{E}_{Y \sim \mathsf{Z}}[(\mathbb{E}[z_1^2 - 1|Y])^2]. \tag{78}$$

Theorem 4.2 in (Troiani et al., 2025) further establishes that there exists an efficient algorithm (in particular an optimal Approximate Message Passing scheme) which can weakly recover what is known as *easy* subspace. The latter, according to Definition 4 in the same work, corresponds to the full $\mathrm{span}(\boldsymbol{W}^\star)$, since there does not exist $\boldsymbol{v} \in \mathbb{R}^p$ such that $\boldsymbol{v}^\top \boldsymbol{G}(y)\boldsymbol{v} = 0$ a.s. over $y$. Hence, by Definition 4.1, for the committee machine with even activation functions

$$\alpha_{\mathrm{WR}}^{\mathrm{Alg.}} = \alpha_{\mathrm{spec.}}^{\mathrm{Alg.}} \tag{79}$$

Note that, for a generic non-even activation[4]

$$\mathbb{E}[z_h|y] = \mathbb{E}[z_k|y] \neq 0, \quad \forall k, h \in [p], \tag{80}$$

due to permutational symmetry. As $\mathbb{E}[\boldsymbol{z}|y] \in \mathrm{span}(\boldsymbol{1}_p)$, the trivial subspace only includes unspecialized estimators, which can therefore be learned at any finite sample complexity $\alpha > 0$.

The proof of Theorem 4.2 follows steps analogous to the one for Theorem 3.2 presented in Appendix A, leading to

$$\lim_{p \to \infty} \frac{p^{\beta_\star}}{\alpha_{\mathrm{WR}}^{\mathrm{Alg.}}} = \frac{\mu_{\beta_\star}^2}{\sqrt{1 + \Delta}\beta_\star!} \tag{81}$$

In particular, denote with $S_p$ the random variable $S_p := \frac{1}{\sqrt{p}} \sum_{i=2}^p \sigma(z_p)$, then $Y \sim \mathcal{N}\left( \frac{1}{\sqrt{p}}\sigma(z_1) + S_p, \Delta \right)$. Without loss of generality, we consider $\mathbb{E}_z[\sigma(z)] = 0$ and $\mathbb{E}_z[\sigma^2(z)] = 1$,[5] therefore $\mathbb{E}[S_p] = 0$ and $\mathbb{E}[S_p^2] = 1 - p^{-1}$. Define the Gaussian density with variance $\Delta$ as $\phi_\Delta(x) := e^{-x^2/(2\Delta)}/\sqrt{2\pi\Delta}$, by Taylor's theorem, for any $\beta \in \mathbb{N}$

$$\phi_\Delta(x + \delta) = \sum_{j=1}^\beta \frac{\phi_\Delta^{(j)}(x)}{\beta!} \delta^j + R_\delta(x) \tag{82}$$

---

[4]We exclude here fine-tuned examples of non-even model with GE = 2.

[5]Such assumptions correspond to a shift and rescaling of the original problem, which do not affect the result of Theorem 4.2.

with $\phi^{(j)}(x) = \frac{\mathrm{d}^j}{\mathrm{d}x^j}\phi(x)$ and $R_\delta(x)$ the remainder term. In some instances, it will be useful to consider the Lagrange form of the remainder $R_\delta(x) = \frac{\delta^{k+1}}{(k+1)!}\phi_\Delta^{(k+1)}(x + \theta\delta)$, for some $\theta \in [0, 1]$. Recalling that, by definition of the NSE $\beta_\star$, $\mu_\beta = 0$ for $\beta < \beta_\star$,

$$Z(y)\mathbb{E}[z_1^2 - 1|y] = (-1)^{\beta_\star}\frac{\lambda^{\beta_\star/2}\mu_{\beta_\star}}{\beta_\star!}\mathbb{E}_{S_p}[\phi_\Delta^{(\beta_\star)}(y - S_p)] + \mathbb{E}_{z,S_p}[(z^2 - 1)R_{\sigma(z)/\sqrt{p}}(y - S_p)], \tag{83}$$

with $\mu_\beta := \mathbb{E}_z[\sigma^\beta(z)(z^2 - 1)]$. Then,

$$\left|(\alpha_{\mathrm{WR}}^{\mathrm{Alg.}})^{-1} - p^{-\beta_\star}\frac{\mu_{\beta_\star}^2}{\beta_\star!^2}\int \mathrm{d}y \frac{1}{Z(y)}(\mathbb{E}_{S_p}[\phi_\Delta^{(\beta_\star)}(y - S_p)])^2\right| \tag{84}$$

$$\leq \int \mathrm{d}y \frac{\left(\mathbb{E}_{z,S_p}[(z^2 - 1)R_{\sigma(z)/\sqrt{p}}(y - S_p)]\right)^2}{Z(y)} \qquad \text{(by the triangle ineq.)}$$

$$= \int \mathrm{d}y \frac{\left(\mathbb{E}_{z,S_p}[(z^2 - 1)R_{\sigma(z)/\sqrt{p}}(y - S_p)\sqrt{\frac{Z(y)}{Z(y)}}]\right)^2}{Z(y)}$$

$$\leq \int \mathrm{d}y \mathbb{E}_{S_p}\left[\frac{\left(\mathbb{E}_z[(z^2 - 1)R_{\sigma(z)/\sqrt{p}}(y - S_p)\sqrt{\frac{\mathbb{E}_z[\phi_\Delta(y - \sigma(z)/\sqrt{p} - S_p)]}{\mathbb{E}_z[\phi_\Delta(y - \sigma(z)/\sqrt{p} - S_p)]}}]\right)^2}{Z(y)}\right] \qquad \text{(by Cauchy-Schwarz ineq.)}$$

$$= \int \mathrm{d}y \mathbb{E}_{S_p}\left[\frac{\left(\mathbb{E}_z[(z^2 - 1)R_{\sigma(z)/\sqrt{p}}(y - S_p)]\right)^2}{\mathbb{E}_z[\phi_\Delta(y - \sigma(z)/\sqrt{p} - S_p)]}\right]$$

$$= \int \mathrm{d}y \frac{\left(\mathbb{E}_z[(z^2 - 1)R_{\sigma(z)/\sqrt{p}}(y)]\right)^2}{\mathbb{E}_z[\phi_\Delta(y - \sigma(z)/\sqrt{p})]}$$

$$\leq \int \mathrm{d}y \frac{\mathbb{E}_z\left[([(z^2 - 1)R_{\sigma(z)/\sqrt{p}}(y)])^2\right]}{\mathbb{E}_z[\phi_\Delta(y - \sigma(z)/\sqrt{p})]} \qquad \text{(by Jensen's ineq.)} \tag{85}$$

In (84), we characterize $\lim_{p\to\infty}\int \mathrm{d}y \frac{1}{Z(y)}(\mathbb{E}_{S_p}\phi_\Delta^{(\beta_\star)}(y - S_p))^2$. For $M > 0$, define the indicator function $\mathbf{1}_M(z) := 1$ if $|\sigma(z)| < M$ and $\mathbf{1}_M(z) := 0$ otherwise. It is straightforward to show that there exists a constant $M$ such that $\Pr(|\sigma(z)| < M) > 0$. For such choice of $M$, for all $p > M^2/\Delta$

$$Z(y) = \mathbb{E}_{z,S_p}\left[\phi_\Delta(y - p^{-1/2}\sigma(z) - S_p)\right] \tag{86}$$

$$\geq \mathbb{E}_{S_p}[\phi_\Delta(y - S_p)e^{-|y-S_p|/\sqrt{\Delta}}]\underbrace{\frac{\Pr(|\sigma(z)| < M)}{\sqrt{e}}}_{:=C_M}, \qquad \text{(by eq. (23))} \tag{87}$$

and consequently

$$\frac{1}{\mathsf{Z}(y)}(\mathbb{E}_{S_p}[\phi_\Delta^{(\beta_\star)}(y-S_p)])^2 \tag{88}$$

$$\leq C_M^{-1}\frac{(\mathbb{E}_{S_p}[\phi_\Delta^{(\beta_\star)}(y-S_p)])^2}{\mathbb{E}_{S_p}[\phi_\Delta(y-S_p)e^{-|y-S_p|/\sqrt{\Delta}}]} \tag{89}$$

$$=C_M^{-1}\frac{\left(\mathbb{E}_{S_p}\left[\phi_\Delta^{(\beta_\star)}(y-S_p)\sqrt{\frac{\phi_\Delta(y-S_p)e^{-|y-S_p|/\sqrt{\Delta}}}{\phi_\Delta(y-S_p)e^{-|y-S_p|/\sqrt{\Delta}}}}\right]\right)^2}{\mathbb{E}_{S_p}[\phi_\Delta(y-S_p)e^{-|y-S_p|/\sqrt{\Delta}}]} \tag{90}$$

$$\leq C_M^{-1}\frac{\mathbb{E}_{S_p}\left[e^{+|y-S_p|/\sqrt{\Delta}}\frac{(\phi_\Delta^{(\beta_\star)}(y-S_p))^2}{\phi_\Delta(y-S_p)}\right]}{\mathbb{E}_{S_p}[\phi_\Delta(y-S_p)e^{-|y-S_p|/\sqrt{\Delta}}]}\mathbb{E}_{S_p}[\phi_\Delta(y-S_p)e^{-|y-S_p|/\sqrt{\Delta}}] \qquad \text{(by Cauchy-Schwarz ineq.)} \tag{91}$$

$$=(\Delta C_M)^{-1}\mathbb{E}_{S_p}\left[\phi_\Delta(y-S_p)e^{|y-S_p|}\mathrm{He}_{\beta_\star}^2\left(\frac{y-S_p}{\sqrt{\Delta}}\right)\right]=:G_p(y). \tag{92}$$

Note that $\int \mathrm{d}y G_p(y)$ is finite and independent of $p$,

$$\int \mathrm{d}y G_p(y) = \frac{1}{C_M\sqrt{\Delta}}\int \mathrm{d}y \phi_1(y)e^{+|y|}\mathrm{He}_{\beta_\star}^2(y) < \infty. \tag{93}$$

Therefore, by Pratt's Lemma (Pratt, 1960),

$$\lim_{p\to\infty}\int \mathrm{d}y\frac{(\mathbb{E}_{S_p}[\phi_\Delta^{(\beta_\star)}(y-S_p)])^2}{\mathsf{Z}(y)} = \int \mathrm{d}y \lim_{p\to\infty}\frac{(\mathbb{E}_{S_p}[\phi_\Delta^{(\beta_\star)}(y-S_p)])^2}{\mathbb{E}_{z,S_p}[\phi_\Delta(y-\frac{1}{\sqrt{p}}\sigma(z)-S_p)]}. \tag{94}$$

In the limit $p \to \infty$, the central limit theorem implies that the variables $S_p$ and $(p^{-1/2}\sigma(z)+S_p)$ converge in distribution to $\mathcal{N}(0,1)$, and, recalling that $\mathbb{E}_{Z\sim\mathcal{N}(0,1)}[\phi_\Delta(x-Z)] = \phi_{\Delta+1}(x)$,

$$\int \mathrm{d}y \lim_{p\to\infty}\frac{(\mathbb{E}_{S_p}[\phi_\Delta^{(\beta_\star)}(y-S_p)])^2}{\mathbb{E}_{z,S_p}[\phi_\Delta(y-\frac{1}{\sqrt{p}}\sigma(z)-S_p)]} = \frac{1}{1+\Delta}\int \mathrm{d}y\,\phi_{1+\Delta}(y)\mathrm{He}_{\beta_\star}^2\left(\frac{y}{\sqrt{1+\Delta}}\right) \tag{95}$$

$$= \frac{\beta_\star!}{\sqrt{1+\Delta}} \tag{96}$$

In order to complete the proof, we have to show that the remainder contribution in eq. (85) is $o(p^{\beta_\star})$. This directly follows by noticing that such a quantity is equivalent to the one in eq. (22) for single-index models. Substituting $\lambda = (p\Delta)^{-1}$, we have shown that this contribution is $o(\lambda^{-\beta_\star})$, yielding the result.

## E. Derivation of Result 4.3

### E.1. Setting and previous results

Recall that we consider the setting of eq. (5):

$$\left\{y_i = \frac{1}{\sqrt{p}}\sum_{k=1}^{p}\sigma(\boldsymbol{w}_k^\star \cdot \boldsymbol{x}_i)\right\}_{i=1}^{n}. \tag{97}$$

We assume Gaussian prior $\boldsymbol{w}_k^\star \overset{\text{i.i.d.}}{\sim} \mathcal{N}(0, \mathrm{I}_d/d)$ and Gaussian data $\boldsymbol{x}_i \overset{\text{i.i.d.}}{\sim} \mathcal{N}(0, \mathrm{I}_d)$. Without loss of generality, we assume that $\sigma$ has zero projection on the first Hermite polynomial:

$$\mathbb{E}[\varphi(G)] = 0, \tag{98}$$

for $G \sim \mathcal{N}(0,1)$. Recall finally that we are in the regime of $n = \Theta(d)$ data samples, with $n/d \to \alpha$, and $p = \mathcal{O}(1)$ as $d \to \infty$. Let us summarize the known main results for the optimal errors achievable in this setting, as given in (Aubin et al., 2019). We denote $\mathcal{S}_p^+$ the set of $p \times p$ real symmetric matrices, which are positive semidefinite.

We define, for $\boldsymbol{z} \in \mathbb{R}^p$:

$$A(\boldsymbol{z}) := \frac{1}{\sqrt{p}} \sum_{k=1}^{p} \sigma(z_k). \tag{99}$$

For $\boldsymbol{q} \in \mathcal{S}_p^+$ such that $\boldsymbol{q} \preceq \mathrm{I}_p$ , and any $y \in \mathbb{R}, \boldsymbol{\xi} \in \mathbb{R}^p$, we let:

$$\begin{cases} I(y, \boldsymbol{\xi}; \boldsymbol{q}) & := \mathbb{E}_{\boldsymbol{Z} \sim \mathcal{N}(0, \mathrm{I}_p)} \left[ \delta \left( y - A \left[ (\mathrm{I}_p - \boldsymbol{q})^{1/2} \boldsymbol{Z} + \boldsymbol{q}^{1/2} \boldsymbol{\xi} \right] \right) \right], \\ \Psi(\boldsymbol{q}) & := \int \mathrm{d}y \, \mathbb{E}_{\boldsymbol{\xi} \sim \mathcal{N}(0, \mathrm{I}_p)} \left[ I(y, \boldsymbol{\xi}; \boldsymbol{q}) \log I(y, \boldsymbol{\xi}; \boldsymbol{q}) \right]. \end{cases} \tag{100}$$

From there we define the so-called *replica-symmetric potential*

$$f(\alpha; \boldsymbol{q}) := \frac{1}{2} \mathrm{Tr}[\boldsymbol{q}] + \frac{1}{2} \log \det[\mathrm{I}_p - \boldsymbol{q}] + \alpha \Psi(\boldsymbol{q}). \tag{101}$$

Notice that $f(\alpha; \boldsymbol{q})$ is invariant under permutation symmetry of the indices: more precisely if $\boldsymbol{P} \in \mathbb{R}^{p \times p}$ is a permutation matrix, then $f(\alpha; \boldsymbol{q}) = f(\alpha; \boldsymbol{P}^\top \boldsymbol{q} \boldsymbol{P})$. This follows from permutation invariance of $A(\boldsymbol{z})$ and rotation invariance of the Gaussian distribution.

### E.1.1. INFORMATION-THEORETIC LIMITS

At the information-theoretic / statistical level, the optimal recovery of $\boldsymbol{W}^\star$ can be characterized in the high-dimensional limit by the properties of $f(\alpha; \boldsymbol{q})$. This was shown in series of works in physics and computer science, which we can summarize in the following statement.

**Theorem E.1** ((Aubin et al., 2019; Barbier & Reeves, 2020; Reeves, 2020))**.** *Let*

$$\boldsymbol{q}^\star := \underset{0 \preceq \boldsymbol{q} \preceq \mathrm{I}_p}{\arg\max} f(\alpha; \boldsymbol{q}). \tag{102}$$

*Denote* $\hat{\mathbf{W}}_{\mathrm{opt}}(\mathcal{D}) := \mathbb{E}[\boldsymbol{W}^\star | \mathcal{D}]$ *the Bayes-optimal estimator of* $\boldsymbol{W}^\star$ *given the dataset* $\mathcal{D} = \{\boldsymbol{x}_i, y_i\}_{i=1}^n$ *in eq. (97). Under a technical assumption (see the remark below), and as* $n, d \to \infty$ *with* $n/d \to \alpha$*, we have*

$$\hat{\mathbf{W}}_{\mathrm{opt}}(\hat{\mathbf{W}}_{\mathrm{opt}})^\top \overset{\mathrm{P}}{\to} \boldsymbol{q}^\star, \;\; \text{and} \;\; \hat{\mathbf{W}}_{\mathrm{opt}}(\boldsymbol{W}^\star)^\top \overset{\mathrm{P}}{\to} \boldsymbol{q}^\star,$$

*where* $\overset{\mathrm{P}}{\to}$ *denotes convergence in probability (with respect to the randomness of all* $(\{\boldsymbol{x}_i, y_i\}, \boldsymbol{W}^\star)$*). In particular the asymptotic minimal mean-squared error on the estimation of* $\boldsymbol{W}^\star$ *satisfies:*

$$\min_{\hat{\mathbf{W}}(\mathcal{D})} \|\hat{\mathbf{W}} - \boldsymbol{W}^\star\|_F^2 = \|\hat{\mathbf{W}}_{\mathrm{opt}} - \boldsymbol{W}^\star\|_F^2 \overset{\mathrm{P}}{\to} \mathrm{Tr}[\mathrm{I}_p - \boldsymbol{q}^\star].$$

**Remark –** We note that, on a technical level, and as discussed e.g. in (Aubin et al., 2019), Theorem E.1 requires adding an infinitesimal amount of side information on $\boldsymbol{W}^\star$ to the observations, allowing e.g. to break symmetries of the model, exactly like in single-index models (Barbier et al., 2019). Since we do not state Result 4.3 as a theorem (as we will in any case rely on assumptions later on), we do not discuss this technical point further, and essentially assume this infinitesimal side-information to be present.

### E.1.2. COMPUTATIONAL LIMITS

Our main tool to characterize the computational thresholds in multi-index models is the performance of the *Approximate Message-Passing* (AMP, sometimes Bayes-AMP) algorithm. We refer to (Aubin et al., 2019; Troiani et al., 2025) for a precise definition of the iterations of the AMP algorithm in general multi-index models (including the one of eq. (5)), and to the main text for a discussion of the optimality of AMP in terms of mean-squared error among first-order methods..

Rather than the precise form of the AMP iterations, the main result we need here is a deterministic characterization of their limiting behavior in the high-dimensional limit, a result known as *state evolution*. In multi-index models, state evolution was derived in (Aubin et al., 2019), and later proven in (Gerbelot & Berthier, 2023). We state it here informally.

**Proposition E.2** (State evolution of AMP, informal (Aubin et al., 2019; Gerbelot & Berthier, 2023)). *Consider the AMP algorithm initialized in $\hat{\mathbf{W}}^0$. For any $t \geq 0$, denote $\hat{\mathbf{W}}^t \in \mathbb{R}^{p \times d}$ the state of the AMP algorithm after $t$ iterations. In the high-dimensional limit $n, d \to \infty$ with $n/d \to \alpha$, we have for any $t \geq 1$:*

$$\hat{\mathbf{W}}^t(\hat{\mathbf{W}}^t)^T \overset{\mathrm{P}}{\to} \boldsymbol{q}^t, \ \ and \ \ \hat{\mathbf{W}}^t(\boldsymbol{W}^\star)^T \overset{\mathrm{P}}{\to} \boldsymbol{q}^t,$$

*where $\overset{\mathrm{P}}{\to}$ denotes convergence in probability (with respect to the randomness of all $(\{\boldsymbol{x}_i, y_i\}, \boldsymbol{W}^\star, \hat{\mathbf{W}}^0)$). Here we assumed that $\hat{\mathbf{W}}^0(\boldsymbol{W}^\star)^T \to \boldsymbol{q}^0$, and we have that $\boldsymbol{q}^t$ satisfies the deterministic recursion, for any $t \geq 0$:*

$$\boldsymbol{q}^{t+1} = G\left[2\alpha\nabla\Psi(\boldsymbol{q}^t)\right], \tag{103}$$

*where $G(\boldsymbol{M}) := (\mathrm{I}_p + \boldsymbol{M})^{-1}\boldsymbol{M}$. Equivalently:*

$$\boldsymbol{q}^{t+1}[\mathrm{I}_p - \boldsymbol{q}^{t+1}] = 2\alpha\nabla\Psi(\boldsymbol{q}^t). \tag{104}$$

**Remark** – Beyond some technical conditions, the main "imprecision" in Proposition E.2 is the discussion of the possible initialization point $\hat{\mathbf{W}}^0$. In general, as in the information-theoretic results above, we implicitly assume that we have access to an infinitesimal amount of side-information, which allows to initialize AMP in a very small but non-zero $\boldsymbol{q}^0$. This allows for example to probe weak recovery by considering the stability of the point $\boldsymbol{q} = 0$ under the state-evolution iterations of eq. (103),(104) (e.g. weak recovery may be possible even if $\boldsymbol{q} = 0$ is a fixed point, if it is unstable). We refer to (Mondelli & Montanari, 2019; Barbier et al., 2019; Maillard et al., 2020a; Troiani et al., 2025) for more discussion on this technical point. Similarly, we will here probe specialization by considering the stability of the whole unspecialized subspace $\mathrm{span}(\mathbf{1}_p\mathbf{1}_p^T)$ under these iterations.

Notice that eq. (103) can be seen as an iterative scheme that, starting from $\boldsymbol{q}^0$, attempts to find a zero of the derivative $\nabla f(\alpha; \boldsymbol{q})$ in eq. (101). Informally, the results above actually allow to characterize both the information-theoretic and computational limits through the single function $f(\alpha; \boldsymbol{q})$.

We now turn to expanding the results of Theorem E.1 and Proposition E.2 to leading order as $p \to \infty$.

## E.2. Committee symmetry

In order to expand the results above as $p \to \infty$, we make the following hypothesis: both $\boldsymbol{q}^\star$ in eq. (102) and $\boldsymbol{q}^t$ in eq. (103) *do not spontaneously break* the permutation symmetry described above (below eq. (101)). This assumption is sometimes called *committee symmetry* in the statistical physics literature), see e.g. (Schwarze, 1993; Aubin et al., 2019; Barbier et al., 2025; Citton et al., 2025). Concretely, such a matrix $\boldsymbol{q}$ can be written as:

$$\boldsymbol{q} = q_d\mathrm{I}_p + \frac{q_a}{p}\mathbf{1}_p\mathbf{1}_p^\top. \tag{105}$$

Notice that by the constraint $\mathrm{I}_p \succeq \boldsymbol{q} \succeq 0$, we have $0 \leq q_d \leq 1$ and $0 \leq q_a + q_d \leq 1$. While eq. (105) posits a global permutation symmetry of the matrix $\boldsymbol{q}$, it allows for two distinct types of solutions:

(i) When $q_d = 0$, the solution considered is *unspecialized*: all the learned weights $\hat{\boldsymbol{w}}_i$ are invariant under permutations of $(\boldsymbol{w}_1^\star, \cdots, \boldsymbol{w}_p^\star)$, and $\boldsymbol{q} \in \mathrm{span}(\mathbf{1}_p\mathbf{1}_p^T)$. There, the solution can be attained by simple linear regression, see eq. (10).

(ii) When $q_d > 0$, this symmetry is broken, and the solution is *specialized*. Each learned weight $\hat{\boldsymbol{w}}_i$ aligns with a corresponding "neuron" $\boldsymbol{w}_i^\star$ of the teacher.

Let us now use the form of eq. (105) in eqs. (102) and (103). The first term of $f(\alpha; \boldsymbol{q})$ is easy to compute:

$$\frac{1}{2}\mathrm{Tr}[\boldsymbol{q}] + \frac{1}{2}\log\det[\mathrm{I}_p - \boldsymbol{q}] = \frac{q_a + pq_d}{2} + \frac{p-1}{2}\log(1 - q_d) + \frac{1}{2}\log(1 - q_a - q_d). \tag{106}$$

Recall the definition of $\Psi$ in eq. (100). We reach the following simplification in the large-$p$ limit, for any committe-symmetric $\boldsymbol{q}$:

$$\Psi(\boldsymbol{q}) = C - \frac{1}{2}\log\left[\gamma_2(q_d) - q_a(\mathbb{E}_G[G\sigma(G)])^2\right] + \mathcal{O}(p^{-1/2}), \tag{107}$$

with $C \in \mathbb{R}$ a constant (independent of $\boldsymbol{q}$), $G \sim \mathcal{N}(0, 1)$, and

$$\gamma_2(q_d) := \mathbb{E}_G[\sigma(G)^2] - \mathbb{E}_{(x,y)\sim\mathcal{N}(0,Q_d)}\left[\sigma(x)\,\sigma(y)\right], \quad Q_d := \begin{pmatrix} 1 & q_d \\ q_d & 1 \end{pmatrix}. \tag{108}$$

**E.3. Large-$p$ expansion of $\Psi(q)$**

We now detail how to obtain eq. (107). Recall eq. (100): $I(y, \boldsymbol{\xi}; \boldsymbol{q})$ is the PDF of the random variable $y = A(\mathbf{X})$, with $\mathbf{X} := (\mathrm{I}_p - \boldsymbol{q})^{1/2}\mathbf{Z} + \boldsymbol{q}^{1/2}\boldsymbol{\xi}$ a Gaussian vector with mean $\boldsymbol{q}^{1/2}\boldsymbol{\xi}$ and covariance $(\mathrm{I}_p - \boldsymbol{q})$. As $p \to \infty$, it is natural to expect that the distribution of $y$ approaches the one of a Gaussian random variable. We make this intuition more formal (and derive the mean and variance of the variable) by considering the characteristic function of $y$, i.e. the Fourier transform of $I(y, \boldsymbol{\xi}; \boldsymbol{q})$. As we do not state Result 4.3 as a theorem (and it relies in any case on the unproven committee symmetry assumption), we present the computation of this characteristic function at a non-rigorous level[6], and leave a mathematical proof for future work. We note that this derivation is similar to computations contained in (Aubin et al., 2019; Barbier et al., 2025), nevertheless we write them in full details for clarity of the presentation.

$$\varphi(\hat{u}, \boldsymbol{\xi}; \boldsymbol{q}) := \mathbb{E}_y[\exp(-i\hat{u}y)] = \int_{\mathbb{R}^p} \frac{\mathrm{d}\mathbf{X}}{(2\pi)^{p/2}\sqrt{\det(\mathrm{I}_p - \boldsymbol{q})}} e^{-\frac{1}{2}(\mathbf{X} - \boldsymbol{q}^{1/2}\boldsymbol{\xi})^\top (\mathrm{I}_p - \boldsymbol{q})^{-1}(\mathbf{X} - \boldsymbol{q}^{1/2}\boldsymbol{\xi}) - \frac{i\hat{u}}{\sqrt{p}}\sum_{k=1}^p \sigma(X_k)}.$$

Using the committee symmetry assumption (eq. (105)), and expanding the product inside the exponential, we get:

$$\varphi(\hat{u}, \boldsymbol{\xi}; \boldsymbol{q}) = \frac{e^{-\frac{\boldsymbol{\xi}^\top \boldsymbol{q}(\mathrm{I}_p - \boldsymbol{q})^{-1}\boldsymbol{\xi}}{2} - \frac{1}{2}\log\frac{1 - q_a - q_d}{1 - q_d}}}{(2\pi(1 - q_d))^{p/2}} \int_{\mathbb{R}^p} \mathrm{d}\mathbf{X}\, e^{-\frac{\|\mathbf{X}\|^2}{2(1 - q_d)} - \frac{q_a}{2p(1 - q_d)(1 - q_a - q_d)}(\mathbf{1}_p^\top \mathbf{X})^2 + (\boldsymbol{v}_{\boldsymbol{\xi}}^\top \mathbf{X}) - \frac{i\hat{u}}{\sqrt{p}}\sum_{k=1}^p \sigma(X_k)}.$$

Here we defined $\boldsymbol{v}_{\boldsymbol{\xi}} := (\mathrm{I}_p - \boldsymbol{q})^{-1}\boldsymbol{q}^{1/2}\boldsymbol{\xi}$ and $\mathbf{1}_p$ is the all-ones vector. We now factorize the integral over $\mathbf{X}$ by introducing an extra variable $w := (\mathbf{1}_m^\top \mathbf{X})/\sqrt{p}$:

$$\varphi(\hat{u}, \boldsymbol{\xi}; \boldsymbol{q}) = \frac{e^{-\frac{\boldsymbol{\xi}^\top \boldsymbol{q}(\mathrm{I}_p - \boldsymbol{q})^{-1}\boldsymbol{\xi}}{2} - \frac{1}{2}\log\frac{1 - q_a - q_d}{1 - q_d}}}{(2\pi(1 - q_d))^{p/2}} \int \frac{\mathrm{d}w\,\mathrm{d}\hat{w}}{2\pi} e^{iw\hat{w} - \frac{q_a}{2(1 - q_d)(1 - q_a - q_d)}w^2}$$

$$\times \int_{\mathbb{R}^p} \mathrm{d}\mathbf{X}\, e^{-\frac{\|\mathbf{X}\|^2}{2(1 - q_d)} - \frac{i\hat{w}\mathbf{1}_p^\top \mathbf{X}}{\sqrt{p}} + (\boldsymbol{v}_{\boldsymbol{\xi}}^\top \mathbf{X}) - \frac{i\hat{u}}{\sqrt{p}}\sum_{k=1}^p \sigma(X_k)},$$

$$= e^{-\frac{\boldsymbol{\xi}^\top \boldsymbol{q}(\mathrm{I}_p - \boldsymbol{q})^{-1}\boldsymbol{\xi}}{2} - \frac{1}{2}\log\frac{1 - q_a - q_d}{1 - q_d}} \int \frac{\mathrm{d}w\,\mathrm{d}\hat{w}}{2\pi} e^{iw\hat{w} - \frac{q_a}{2(1 - q_d)(1 - q_a - q_d)}w^2}$$

$$\times \prod_{k=1}^p \int \frac{\mathrm{d}x}{(2\pi(1 - q_d))^{1/2}} e^{-\frac{x^2}{2(1 - q_d)} - \frac{i\hat{w}x}{\sqrt{p}} + v_k x - \frac{i\hat{u}}{\sqrt{m}}\sigma(x)}, \tag{109}$$

with $(v_k)_{k=1}^p$ the entries of $\boldsymbol{v}_{\boldsymbol{\xi}}$. We denote $J_k(\boldsymbol{\xi}, \hat{w}, \hat{u})$ the integral appearing in the last product

$$J_k(\boldsymbol{\xi}, \hat{w}, \hat{u}) := \int \frac{\mathrm{d}x}{(2\pi(1 - q_d))^{1/2}} e^{-\frac{x^2}{2(1 - q_d)} - \frac{i\hat{w}x}{\sqrt{p}} + v_k x - \frac{i\hat{u}}{\sqrt{p}}\sigma(x)}.$$

We now focus on the large-$p$ expansion of $\sum_{k=1}^p \log J_k(\boldsymbol{\xi}, \hat{w}, \hat{u})$, which appears in eq. (109). Notice that, from eq. (105):

$$\boldsymbol{v}_{\boldsymbol{\xi}} = (\mathrm{I}_p - \boldsymbol{q})^{-1}\boldsymbol{q}^{1/2}\boldsymbol{\xi} = \frac{\sqrt{q_d}}{1 - q_d}\boldsymbol{\xi} + \underbrace{\left[\frac{1}{2}\left(\frac{1}{1 + \sqrt{q_d}} - \frac{1}{1 - \sqrt{q_d}} + \frac{2\sqrt{q_a + q_d}}{1 - q_a - q_d}\right)\right]}_{=:h(q_d, q_a)} \frac{(\boldsymbol{\xi}^\top \mathbf{1}_p)}{p}\mathbf{1}_p. \tag{110}$$

Therefore we have, denoting $s(\boldsymbol{\xi}) := (\mathbf{1}_p^\top \boldsymbol{\xi})/\sqrt{p}$ (which is $\Theta(1)$ as $p \to \infty$):

$$J_k(\boldsymbol{\xi}, \hat{w}, \hat{u}) = \int \frac{\mathrm{d}x}{(2\pi(1 - q_d))^{1/2}} e^{-\frac{x^2}{2(1 - q_d)} - \frac{i\hat{w}x}{\sqrt{p}} + \frac{\sqrt{q_d}}{1 - q_d}\xi_k x + \frac{hs(\boldsymbol{\xi})x}{\sqrt{p}} - \frac{i\hat{u}}{\sqrt{p}}\sigma(x)}.$$

Expanding the terms inside the integral that are $\mathcal{O}(1/\sqrt{p})$, we get:

$$J_k(\boldsymbol{\xi}, \hat{w}, \hat{u}) = \int \frac{\mathrm{d}x}{(2\pi(1 - q_d))^{1/2}} e^{-\frac{x^2}{2(1 - q_d)} + \frac{\sqrt{q_d}}{1 - q_d}\xi_k x} \left[1 + \frac{(hs(\boldsymbol{\xi}) - i\hat{w})x}{\sqrt{p}} - \frac{i\hat{u}}{\sqrt{p}}\sigma(x)\right.$$

$$\left. + \frac{1}{2}\left(\frac{(hs(\boldsymbol{\xi}) - i\hat{w})x}{\sqrt{p}} - \frac{i\hat{u}}{\sqrt{p}}\sigma(x)\right)^2 + \mathcal{O}(p^{-3/2})\right]. \tag{111}$$

---

[6]While not mathematically proven, our derivation does not make use of any heuristic method, a proof would rather require a more precise control of the error terms.

Let us denote, for $\gamma > 0$, $\beta \in \mathbb{R}$, and $l, m \in \mathbb{N}$:

$$F_{ml}(\gamma, \beta) := \frac{e^{-\frac{\beta^2}{2\gamma}}}{\sqrt{2\pi/\gamma}} \int \mathrm{d}x e^{-\frac{\gamma}{2}x^2 + \beta x} x^m \sigma(x)^l = \langle x^m \sigma(x)^l \rangle_{\gamma, \beta}, \tag{112}$$

where $\langle \cdot \rangle_{\gamma, \beta}$ is the Gaussian measure with weight $e^{-\frac{\gamma}{2}x^2 + \beta x}$. Notice that $F_{00} = 1$, $F_{10}(\gamma, \beta) = \beta/\gamma$, and $F_{20}(\gamma, \beta) = \gamma^{-1} + \beta^2/\gamma^2$. Denote

$$\begin{cases} \gamma & := \dfrac{1}{1 - q_d}, \\ \beta_k & := \dfrac{\sqrt{q_d}}{1 - q_d}\xi_k. \end{cases} \tag{113}$$

Going back to eq. (111), we get (denoting $F_{ml}$ for $F_{ml}(\gamma, \beta_k)$ for lightness):

$$J_k(\boldsymbol{\xi}, \hat{w}, \hat{u}) = e^{\frac{q_d \xi_k^2}{2(1 - q_d)}} \left[ 1 + \frac{(hs(\boldsymbol{\xi}) - i\hat{w})F_{10}}{\sqrt{p}} - \frac{i\hat{u}F_{01}}{\sqrt{p}} + \frac{(hs(\boldsymbol{\xi}) - i\hat{w})^2 F_{20} - \hat{u}^2 F_{02}}{2p} \right.$$
$$\left. - \frac{i\hat{u}(hs(\boldsymbol{\xi}) - i\hat{w})F_{11}}{p} + \mathcal{O}(p^{-3/2}) \right].$$

Therefore:

$$\log J_k(\boldsymbol{\xi}, \hat{w}, \hat{u}) = \frac{q_d \xi_k^2}{2(1 - q_d)} + \log \left[ 1 + \frac{(hs(\boldsymbol{\xi}) - i\hat{w})F_{10}}{\sqrt{p}} - \frac{i\hat{u}F_{01}}{\sqrt{p}} + \frac{(hs(\boldsymbol{\xi}) - i\hat{w})^2 F_{20} - \hat{u}^2 F_{02}}{2p} \right.$$
$$\left. - \frac{i\hat{u}(hs(\boldsymbol{\xi}) - i\hat{w})F_{11}}{p} + \mathcal{O}(p^{-3/2}) \right],$$
$$= \frac{q_d \xi_k^2}{2(1 - q_d)} + \frac{(hs(\boldsymbol{\xi}) - i\hat{w})F_{10}}{\sqrt{p}} - \frac{i\hat{u}F_{01}}{\sqrt{p}} + \frac{(hs(\boldsymbol{\xi}) - i\hat{w})^2 F_{20} - \hat{u}^2 F_{02}}{2p}$$
$$- \frac{i\hat{u}(hs(\boldsymbol{\xi}) - i\hat{w})F_{11}}{p} - \frac{((hs(\boldsymbol{\xi}) - i\hat{w})F_{10} - i\hat{u}F_{01})^2}{2p} + \mathcal{O}(p^{-3/2}),$$
$$= \frac{q_d \xi_k^2}{2(1 - q_d)} + \frac{(hs(\boldsymbol{\xi}) - i\hat{w})F_{10}}{\sqrt{p}} - \frac{i\hat{u}F_{01}}{\sqrt{p}} - \frac{i\hat{u}(hs(\boldsymbol{\xi}) - i\hat{w})(F_{11} - F_{01}F_{10})}{p}$$
$$+ \frac{(hs(\boldsymbol{\xi}) - i\hat{w})^2(F_{20} - F_{10}^2) - \hat{u}^2(F_{02} - F_{01}^2)}{2p} + \mathcal{O}(p^{-3/2}). \tag{114}$$

Notice that $F_{20} - F_{10}^2 = \gamma^{-1} = 1 - q_d$. Moreover $F_{10} = \beta/\gamma = \sqrt{q_d}\xi_k$. When computing $\sum_{k=1}^p \log J_k(\boldsymbol{\xi}, \hat{w}, \hat{u})$, it will be important to see how the different variables scale. For this, we define the different quantities, that all depend only on the *empirical density of $\boldsymbol{\xi}$*:

$$\begin{cases} s(\boldsymbol{\xi}) & := \dfrac{1}{\sqrt{p}} \displaystyle\sum_{k=1}^p \xi_k, \\ \Gamma_0(\boldsymbol{\xi}) & := \dfrac{1}{\sqrt{p}} \displaystyle\sum_{k=1}^p F_{01}\left(\dfrac{1}{1 - q_d}, \dfrac{\sqrt{q_d}}{1 - q_d}\xi_k\right), \\ \Gamma_1(\boldsymbol{\xi}) & := \dfrac{1}{p} \displaystyle\sum_{k=1}^p [F_{11} - F_{01}F_{10}]\left(\dfrac{1}{1 - q_d}, \dfrac{\sqrt{q_d}}{1 - q_d}\xi_k\right), \\ \Gamma_2(\boldsymbol{\xi}) & := \dfrac{1}{p} \displaystyle\sum_{k=1}^p [F_{02} - F_{01}^2]\left(\dfrac{1}{1 - q_d}, \dfrac{\sqrt{q_d}}{1 - q_d}\xi_k\right). \end{cases} \tag{115}$$

Notice that $\Gamma_1, \Gamma_2$ correspond to the averages of the covariance of $(x, \sigma(x))$ and of the variance of $\sigma(x)$, under the laws $\langle \cdot \rangle_{\gamma, \beta_k}$. We remark that eq. (98) ensures that

$$
\mathbb{E}_{\xi \sim \mathcal{N}(0,1)} \left[ F_{kl} \left( \frac{1}{1 - q_d}, \frac{\sqrt{q_d}}{1 - q_d} \xi \right) \right] = \frac{1}{\sqrt{2\pi(1 - q_d)}} \int \mathrm{d}x e^{-\frac{x^2}{2(1 - q_d)}} x^k \sigma(x)^l \mathbb{E}_\xi \left[ e^{-\frac{q_d \xi^2}{2(1 - q_d)} + \frac{x \sqrt{q_d} \xi}{1 - q_d}} \right],
$$

$$
= \frac{1}{\sqrt{2\pi}} \int \mathrm{d}x e^{-\frac{x^2}{2}} x^k \sigma(x)^l,
$$

$$
= \mathbb{E}_{G \sim \mathcal{N}(0,1)}[G^k \sigma(G)^l]. \tag{116}
$$

In particular, we have $\mathbb{E}_{\xi \sim \mathcal{N}(0,1)} \left[ F_{01} \left( \frac{1}{1 - q_d}, \frac{\sqrt{q_d}}{1 - q_d} \xi \right) \right] = 0$, and by the central limit theorem $\Gamma_0(\boldsymbol{\xi})$ converges to a centered Gaussian random variable as $p \to \infty$. We will come back to this point later on. We reach from eq. (114):

$$
\sum_{k=1}^{p} \log J_k(\boldsymbol{\xi}, \hat{w}, \hat{u}) = \frac{q_d \|\boldsymbol{\xi}\|^2}{2(1 - q_d)} + \sqrt{q_d}(hs(\boldsymbol{\xi}) - i\hat{w})s(\boldsymbol{\xi}) - i\hat{u}\Gamma_0(\boldsymbol{\xi}) - i\hat{u}(hs(\boldsymbol{\xi}) - i\hat{w})\Gamma_1(\boldsymbol{\xi})
$$

$$
+ \frac{(hs(\boldsymbol{\xi}) - i\hat{w})^2(1 - q_d)}{2} - \frac{\hat{u}^2 \Gamma_2(\boldsymbol{\xi})}{2} + \mathcal{O}(p^{-1/2}). \tag{117}
$$

Coming back to eq. (109):

$$
\varphi(\hat{u}, \boldsymbol{\xi}; \boldsymbol{q}) = e^{-\frac{\boldsymbol{\xi}^\top \boldsymbol{q}(\mathrm{I}_p - \boldsymbol{q})^{-1}\boldsymbol{\xi}}{2} - \frac{1}{2}\log\frac{1 - q_a - q_d}{1 - q_d}} \int \frac{\mathrm{d}w\,\mathrm{d}\hat{w}}{2\pi} e^{iw\hat{w} - \frac{q_a}{2(1 - q_d)(1 - q_a - q_d)}w^2}
$$

$$
\times e^{\frac{q_d\|\boldsymbol{\xi}\|^2}{2(1 - q_d)} + \sqrt{q_d}(hs(\boldsymbol{\xi}) - i\hat{w})s(\boldsymbol{\xi}) - i\hat{u}\Gamma_0(\boldsymbol{\xi}) - i\hat{u}(hs(\boldsymbol{\xi}) - i\hat{w})\Gamma_1(\boldsymbol{\xi}) + \frac{(hs(\boldsymbol{\xi}) - i\hat{w})^2(1 - q_d)}{2} - \frac{\hat{u}^2 \Gamma_2(\boldsymbol{\xi})}{2}}[1 + \mathcal{O}(p^{-1/2})]. \tag{118}
$$

Notice that

$$
\boldsymbol{\xi}^\top \boldsymbol{q}(\mathrm{I}_p - \boldsymbol{q})^{-1}\boldsymbol{\xi} = \frac{q_d}{1 - q_d}\|\boldsymbol{\xi}\|^2 + \frac{q_a}{(1 - q_d)(1 - q_a - q_d)}s(\boldsymbol{\xi})^2.
$$

And so we get:

$$
\varphi(\hat{u}, \boldsymbol{\xi}; \boldsymbol{q}) = e^{-\left[\frac{q_a}{(1 - q_d)(1 - q_a - q_d)} - 2h\sqrt{q_d} - (1 - q_d)h^2\right]\frac{s(\boldsymbol{\xi})^2}{2} - \frac{1}{2}\log\frac{1 - q_a - q_d}{1 - q_d}} \int \frac{\mathrm{d}w\,\mathrm{d}\hat{w}}{2\pi} e^{iw\hat{w} - \frac{q_a}{2(1 - q_d)(1 - q_a - q_d)}w^2}
$$

$$
\times e^{-i\sqrt{q_d}\hat{w}s(\boldsymbol{\xi}) - i\hat{u}\Gamma_0(\boldsymbol{\xi}) - i\hat{u}(hs(\boldsymbol{\xi}) - i\hat{w})\Gamma_1(\boldsymbol{\xi}) - \frac{(2ihs(\boldsymbol{\xi})\hat{w} + \hat{w}^2)(1 - q_d)}{2} - \frac{\hat{u}^2 \Gamma_2(\boldsymbol{\xi})}{2}}[1 + \mathcal{O}(p^{-1/2})]. \tag{119}
$$

We now come back to $I(y, \boldsymbol{\xi}; \boldsymbol{q}) = \int \mathrm{d}\hat{u}e^{i\hat{u}y}\varphi(\hat{u}, \boldsymbol{\xi}; \boldsymbol{q})/(2\pi)$. We get:

$$
I(y, \boldsymbol{\xi}; \boldsymbol{q}) = e^{-\left[\frac{q_a}{(1 - q_d)(1 - q_a - q_d)} - 2h\sqrt{q_d} - (1 - q_d)h^2\right]\frac{s(\boldsymbol{\xi})^2}{2} - \frac{1}{2}\log\frac{1 - q_a - q_d}{1 - q_d}} \int \frac{\mathrm{d}\hat{u}\,\mathrm{d}w\,\mathrm{d}\hat{w}}{(2\pi)^2} e^{i\hat{u}y + iw\hat{w} - \frac{q_a}{2(1 - q_d)(1 - q_a - q_d)}w^2}
$$

$$
\times e^{-i\sqrt{q_d}\hat{w}s(\boldsymbol{\xi}) - i\hat{u}\Gamma_0(\boldsymbol{\xi}) - i\hat{u}(hs(\boldsymbol{\xi}) - i\hat{w})\Gamma_1(\boldsymbol{\xi}) - \frac{(2ihs(\boldsymbol{\xi})\hat{w} + \hat{w}^2)(1 - q_d)}{2} - \frac{\hat{u}^2 \Gamma_2(\boldsymbol{\xi})}{2}}[1 + \mathcal{O}(p^{-1/2})]. \tag{120}
$$

The integrals over all $(\hat{u}, w, \hat{w})$ in eq. (120) are Gaussian, so they can be computed easily although the computation is tedious (we drop the dependencies of $s, \{\Gamma_a\}$ on $\boldsymbol{\xi}$ for ligthness). We reach after integration over $(\hat{u}, w)$:

$$
I(y, \boldsymbol{\xi}; \boldsymbol{q}) = \frac{(1 - q_d)}{\sqrt{q_a \Gamma_2}} e^{-\left[\frac{q_a}{(1 - q_d)(1 - q_a - q_d)} - 2h\sqrt{q_d} - (1 - q_d)h^2\right]\frac{s(\boldsymbol{\xi})^2}{2}} \int \frac{\mathrm{d}\hat{w}}{2\pi} e^{-\frac{(1 - q_d)(1 - q_a - q_d)}{2q_a}\hat{w}^2}
$$

$$
\times e^{-i\sqrt{q_d}\hat{w}s(\boldsymbol{\xi}) - \frac{(2ihs(\boldsymbol{\xi})\hat{w} + \hat{w}^2)(1 - q_d)}{2} - \frac{[y - \Gamma_0 - (hs - i\hat{w})\Gamma_1]^2}{2\Gamma_2}}[1 + \mathcal{O}(p^{-1/2})]. \tag{121}
$$

After performing these tedious Gaussian integrals, we obtain the following simple expression:

$$
I(y, \boldsymbol{\xi}) = \frac{1}{\sqrt{2\pi\tau}} e^{-\frac{(y - \mu)^2}{2\tau}} (1 + \mathcal{O}(p^{-1/2})),
$$

$$
\begin{cases}
\tau & := \Gamma_2(\boldsymbol{\xi}) - \dfrac{q_a}{(1 - q_d)^2}\Gamma_1(\boldsymbol{\xi})^2, \\[3mm]
\mu & := \Gamma_0(\boldsymbol{\xi}) + \dfrac{\sqrt{q_a + q_d} - \sqrt{q_d}}{1 - q_d}s(\boldsymbol{\xi})\Gamma_1(\boldsymbol{\xi}).
\end{cases} \tag{122}
$$

**Remark –** This tedious derivation formalized the fact that the random variable $y = F(\mathbf{X})$ is, as $p \to \infty$, approximately Gaussian, and we determined its mean and variance.

Recall the definitions of $s(\boldsymbol{\xi})$ and $\Gamma_a(\boldsymbol{\xi})$ in eq. (115): in particular, these are also functions of $q_d$. Going back to eq. (100), we get:

$$\Psi(\boldsymbol{q}) = \int \mathrm{d}y \, \mathbb{E}_{\boldsymbol{\xi} \sim \mathcal{N}(0, \mathrm{I}_p)} I(y, \boldsymbol{\xi}; \boldsymbol{q}) \log I(y, \boldsymbol{\xi}; \boldsymbol{q}), \tag{123}$$

$$= \mathbb{E}_{\boldsymbol{\xi}} \int \mathrm{d}y \frac{1}{\sqrt{2\pi\tau}} e^{-\frac{y^2}{2\tau}} \left[ -\frac{y^2}{2\tau} - \frac{1}{2} \log 2\pi\tau + \mathcal{O}(p^{-1/2}) \right] \left( 1 + \mathcal{O}(p^{-1/2}) \right), \tag{124}$$

$$= -\frac{1 + \log 2\pi}{2} - \frac{1}{2} \mathbb{E}_{\boldsymbol{\xi}} \log \left[ \Gamma_2(\boldsymbol{\xi}) - \frac{q_a}{(1 - q_d)^2} \Gamma_1(\boldsymbol{\xi})^2 \right] + \mathcal{O}(p^{-1/2}). \tag{125}$$

As we remarked above, by the law of large numbers, $\Gamma_1(\boldsymbol{\xi}), \Gamma_2(\boldsymbol{\xi})$ concentrate on their average as $p \to \infty$, that we denote $\gamma_1(q_d), \gamma_2(q_d)$. They are defined as:

$$\begin{cases} \gamma_1(q_d) &= \mathbb{E}_{\xi \sim \mathcal{N}(0,1)} \left\{ [F_{11} - F_{01}F_{10}] \left( \frac{1}{1 - q_d}, \frac{\sqrt{q_d}}{1 - q_d} \xi \right) \right\}, \\ \gamma_2(q_d) &= \mathbb{E}_{\xi \sim \mathcal{N}(0,1)} \left\{ [F_{02} - F_{01}^2] \left( \frac{1}{1 - q_d}, \frac{\sqrt{q_d}}{1 - q_d} \xi \right) \right\}. \end{cases} \tag{126}$$

Recall the definition of $F_{ml}(\gamma, \beta)$ in eq. (112), and what we showed in eq. (116). We can further show

$$\mathbb{E}_{\xi \sim \mathcal{N}(0,1)} \left[ (F_{kl} \cdot F_{mn}) \left( \frac{1}{1 - q_d}, \frac{\sqrt{q_d}}{1 - q_d} \xi \right) \right]$$

$$= \frac{1}{2\pi(1 - q_d)} \int \mathrm{d}x \, \mathrm{d}y \, e^{-\frac{(x^2 + y^2)}{2(1 - q_d)}} x^k \, y^m \, \sigma(x)^l \, \sigma(y)^n \, \mathbb{E}_\xi \left[ e^{-\frac{q_d \xi^2}{1 - q_d} + \frac{(x+y)\sqrt{q_d}\xi}{1 - q_d}} \right],$$

$$= \frac{1}{2\pi\sqrt{1 - q_d^2}} \int \mathrm{d}x \, \mathrm{d}y \, \exp \left\{ -\frac{1}{2} \begin{pmatrix} x & y \end{pmatrix} \begin{pmatrix} 1 & q_d \\ q_d & 1 \end{pmatrix}^{-1} \begin{pmatrix} x \\ y \end{pmatrix} \right\} x^k \, y^m \, \sigma(x)^l \, \sigma(y)^n,$$

$$= \mathbb{E}_{(x,y) \sim \mathcal{N}(0,Q_d)} \left[ x^k \, y^m \, \sigma(x)^l \, \sigma(y)^n \right]. \tag{127}$$

with $Q_d = \begin{pmatrix} 1 & q_d \\ q_d & 1 \end{pmatrix}$. In particular:

$$\begin{cases} \gamma_1(q_d) &= \mathbb{E}_G[G\sigma(G)] - \mathbb{E}_{(x,y) \sim \mathcal{N}(0,Q_d)} [x \, \sigma(y)] = (1 - q_d) \mathbb{E}_G[G\sigma(G)], \\ \gamma_2(q_d) &= \mathbb{E}_G[\sigma(G)^2] - \mathbb{E}_{(x,y) \sim \mathcal{N}(0,Q_d)} [\sigma(x) \, \sigma(y)]. \end{cases} \tag{128}$$

Recall the Hermite decomposition of $\sigma$ and that we already computed $\gamma_2(q)$, see eq. (58). We get:

$$\begin{cases} \sigma(z) &= \sum_{k=1}^{\infty} \frac{c_k}{k!} \mathrm{He}_k(z), \\ \gamma_2(q) &= \sum_{k=1}^{\infty} \frac{c_k^2}{k!}(1 - q^k) = (1 - q) \sum_{k=1}^{\infty} \frac{c_k^2}{k!} \left( \sum_{l=0}^{k-1} q^l \right) = (1 - q) \sum_{l=0}^{\infty} q^l \left[ \sum_{k=l+1}^{\infty} \frac{c_k^2}{k!} \right]. \end{cases} \tag{129}$$

Plugging eq. (128) in eq. (123) ends our derivation of eq. (107).

### E.4. Statistical and computational limits as $p \to \infty$

Using eq. (107), we now rewrite the conclusions of Theorem E.1 and E.2 in the $p \to \infty$ limit, under the committee-symmetry assumption.

**Information-theoretic limit :** We have $\boldsymbol{q}^\star = q_d \mathrm{I}_p + (q_a/p) \mathbf{1}_p \mathbf{1}_p^\top$. We denote $h := 1 - q_a - q_d$, and $0 \leq q_d, h \leq 1$ correspond to the global maximum of

$$f(\alpha; q_d, h) := \frac{(p-1)q_d}{2} + \frac{p-1}{2} \log(1 - q_d) + \frac{\log h - h}{2} - \frac{\alpha}{2} \left( \log \left[ \gamma_2(q_d) - (1 - q_d)c_1^2 + hc_1^2 \right] + \mathcal{O}(p^{-1/2}) \right). \tag{130}$$

Recall that $c_1 = \mathbb{E}[G\sigma(G)]$. Since $\sigma$ is not a linear function, by eq. (129):

$$\widetilde{\gamma}_2(q) := \gamma_2(q) - (1-q)c_1^2 = \sum_{k=2}^{\infty} \frac{c_k^2}{k!}(1-q^k) = (1-q)\sum_{k=2}^{\infty} \frac{c_k^2}{k!}\left(\sum_{l=0}^{k-1} q^l\right),$$

and the gradient of $f$ is given by:

$$
\begin{cases}
\dfrac{\partial f}{\partial q_d} &= -\dfrac{(p-1)q_d}{2(1-q_d)} - \dfrac{\alpha}{2}\left[\dfrac{\widetilde{\gamma}_2'(q_d)}{\widetilde{\gamma}_2(q_d) + hc_1^2} + \mathcal{O}(p^{-1/2})\right], \\[2ex]
\dfrac{\partial f}{\partial h} &= \dfrac{1-h}{h} - \dfrac{\alpha}{2}\left[\dfrac{c_1^2}{\widetilde{\gamma}_2(q_d) + hc_1^2} + \mathcal{O}(p^{-1/2})\right].
\end{cases}
\tag{131}
$$

**State evolution of AMP:** We have $q^t = q_d^t I_p + (q_a^t/p)\mathbf{1}_p\mathbf{1}_p^\top$. Let us define $h^t = 1 - q_a^t - q_d^t$, and from eq. (104) and the expansion of $\Psi$ in eq. (107), we reach that they satisfy:

$$
\begin{cases}
(p-1)\dfrac{q_d^{t+1}}{1-q_d^{t+1}} &= -\alpha\left[\dfrac{\widetilde{\gamma}_2'(q_d^t)}{\widetilde{\gamma}_2(q_d^t) + c_1^2 h^t} + \mathcal{O}(p^{-1/2})\right], \\[2ex]
\dfrac{1-h^{t+1}}{h^{t+1}} &= \alpha\left[\dfrac{c_1^2}{\widetilde{\gamma}_2(q_d^t) + c_1^2 h^t} + \mathcal{O}(p^{-1/2})\right].
\end{cases}
\tag{132}
$$

We now turn to the specific claims of Result 4.3. We recall that we do not aim here at a fully rigorous treatment, which would need in particular a more precise control of the error terms in eqs. (130) and eq. (132).

### E.4.1. COMPUTATIONAL SPECIALIZATION TRANSITION

We start with the AMP state-evolution equations of eq. (132). As we discussed above, we probe whether specialization is computationally possible by considering the evolution of solutions with a small $q_d^t > 0$ (very small but not going to 0 as $p \to \infty$) under these iterations. On the other hand, we assume an arbitrary $h^t \in (0,1)$, i.e. we do not make any assumption on the "unspecialized" part of the solution.

For $q \ll 1$, we have the expansion

$$\widetilde{\gamma}_2(q) = \mathbb{E}[\sigma(G)^2] - c_1^2 - \frac{c_2^2}{2}q^2 + \mathcal{O}(q^3).$$

Recall that $\mathbb{E}[\sigma(G)^2] - c_1^2 > 0$ since $\sigma$ is non-linear. Linearizing the first equation of eq. (132) at leading order in $q_d$ and $p$ leads to:

$$q_d^{t+1} \simeq \frac{\alpha}{p}\left[\frac{c_2^2 q_d^t}{\mathbb{E}[\sigma(G)^2] - (1-h^t)c_1^2}\right].
\tag{133}$$

Eq. (133) suggests to separate two regimes:

- If $\alpha = o(p)$, then even if the AMP iterates have a small $q_d^t > 0$, they eventually converge to $q_d = o_p(1)$. In this regime, AMP can only recover an *unspecialized* solution.

- If $\alpha = \Theta(p)$, then the situation is slightly more complex. Let $\widetilde{\alpha} := \alpha/p$. If we assume that $c_2 = 0$ (i.e. the NSE $\beta^\star > 1$), then $q_d = o_p(1)$ remains stable under the AMP iterations at least if $\alpha = o(p^{3/2})$, after which the error term $\mathcal{O}(p^{-1/2})$ in eq. (132) might kick in. On the other hand, if $c_2 \neq 0$ ($\beta^\star = 1$), then eq. (133) predicts that $q_d = o_p(1)$ becomes unstable as soon as the following condition is satisfied:

$$\frac{\widetilde{\alpha}c_2^2}{\mathbb{E}[\sigma(G)^2] - (1-h^t)c_1^2} > 1.$$

Since $h^t \in [0,1]$ and $\sigma$ is non-linear (so $\mathbb{E}[\sigma(G)^2] > c_1^2$), the condition

$$\widetilde{\alpha} > \frac{\mathbb{E}[\sigma(G)^2] - c_1^2}{c_2^2} = \sum_{k=2}^{\infty} \frac{c_k^2}{k!c_2^2}$$

implies that AMP develops a specialization transition.

This justifies the claims of Result 4.3 regarding the computational specialization transition.

### E.4.2. INFORMATION-THEORETIC SPECIALIZATION

We focus on eq. (130). Unspecialized estimators correspond to $q_d = o_p(1)$, i.e.

$$f(\alpha; o_p(1), h) = \frac{\log h - h}{2} - \frac{\alpha}{2} \left( \log \left[ \mathbb{E}[\sigma(G)^2] - c_1 + hc_1^2 \right] + o_p(1) \right) + o_p(p).$$

In particular they satisfy that:

$$\frac{1}{p} f(\alpha; o_p(1), h) \le -\frac{1}{2} - \frac{\alpha}{2p} \left( \log \left[ \mathbb{E}[\sigma(G)^2] - c_1 \right] + o_p(1) \right) + o_p(1). \tag{134}$$

We will show that, if $\alpha > p$, there exists a specialized overlap matrix that achieves a larger value of $f$ than the one given by eq. (134). Let $\varepsilon \in (0, 1)$ be small, and define $q_d = 1 - \varepsilon$, $h = \varepsilon$. We have

$$f(\alpha; 1 - \varepsilon, \varepsilon) = \frac{(p-1)(1-\varepsilon)}{2} + \frac{p-1}{2} \log \varepsilon + \frac{\log \varepsilon - \varepsilon}{2} - \frac{\alpha}{2} \left( \log \left[ \gamma_2(1-\varepsilon) \right] + \mathcal{O}(p^{-1/2}) \right).$$

By eq. (129), $\gamma_2(1 - \varepsilon) = \varepsilon L(\varepsilon, \sigma)$, where

$$L(\varepsilon, \sigma) = \sum_{l=0}^{\infty} (1-\varepsilon)^l \sum_{k=l+1}^{\infty} \frac{c_k^2}{k!} \in \left[ \sum_{k=1}^{\infty} \frac{c_k^2}{k!}, \sum_{k=1}^{\infty} \frac{c_k^2}{(k-1)!} \right]. \tag{135}$$

Therefore, if $\widetilde{\alpha} = \alpha/p = \Theta(1)$, the leading order (as $p \to \infty$) of $f(\alpha; 1 - \varepsilon, \varepsilon)$ is given by

$$\frac{1}{p} f(\alpha; 1 - \varepsilon, \varepsilon) \sim \frac{1 - \widetilde{\alpha} \log L(\varepsilon, \sigma)}{2} + \frac{1 - \widetilde{\alpha}}{2} \log \varepsilon + \mathcal{O}(p^{-1/2}).$$

Since $L(\varepsilon, \sigma)$ is bounded away from zero as $\varepsilon \to 0$ by eq. (135), and by eq. (134), if $\widetilde{\alpha} > 1$ it suffices to take $\varepsilon > 0$ small enough to have $f(\alpha; 1 - \varepsilon, \varepsilon) > f(\alpha; o_p(1), h)$.

Notice that for $\varepsilon \downarrow 0$, this solution actually corresponds to a *perfect recovery* of $\boldsymbol{W}^{\star}$. Since we show that $(1/p) f(\alpha; 1 - \varepsilon, \varepsilon) \to +\infty$ as $\varepsilon \downarrow 0$ for $\widetilde{\alpha} > 1$, our computation actually suggests that *perfect recovery* is statistically possible as soon as $\alpha > p(1 + \eta)$ for any finite $\eta > 0$ as $p \to \infty$.

