# OpenReview forum: "A Noise Sensitivity Exponent Controls Large Statistical-to-Computational Gaps in Single- and Multi-Index Models"
_ICML.cc/2026/Conference — ICML 2026 spotlight_

### Official Review · Reviewer_cBtw · 2026-03-07

**Soundness:** 3
**Presentation:** 3
**Significance:** 2
**Originality:** 3
**Overall Recommendation:** 5
**Confidence:** 3

**Summary:**

In this paper, the authors study the statistical-to-computational gap for single-index models with large additive noise and separable multi-index models with Gaussian vector ground-truth weights, in the regime where the ratio of the number of samples and ambient dimension converges to $\alpha > 0$.
They introduce the notion of noise-sensitivity exponent (NSE) of the link function $\sigma$, which is defined to be the smallest $\beta$ such that the second Hermite coefficient of $\sigma^\beta$ is nonzero.
Then, they argue that NSE determines if the learner can efficiently weakly recover the ground-truth weights in above models in the following sense:
* For single-index models with NSE $\beta_*$ and signal-to-noise ratio $\lambda$, the algorithmic weak recovery threshold is $\Theta( \lambda^{-\beta} )$ while the information-theoretic threshold is $\Theta(\lambda^{-1})$.
* For multi-index models with relevant dimension $p \gg 1$, when the link function is even, the algorithmic weak recovery and specialization thresholds are both $\Theta(p^\beta)$.
* They also prove partial results for multi-index models with a non-even link function.

**Compliance With Llm Reviewing Policy:**

Affirmed.

**Final Justification:**

I raised the score from 4 to 5, because the authors explained in the rebuttal why simply removing the linear terms cannot make the learned solution specialized, resolving my main concern.

**Key Questions For Authors:**

See the Weakness section for details.
* Could you explain the technical contribution of this paper over (Barbier et al., 2019) and (Troiani et al., 2025)?
* For multi-index models with a non-even link function, can we remove the influence of the average direction after learning it?

**Limitations:**

Yes.

**Strengths And Weaknesses:**

### Strengths

Overall, this is a decent paper.
In the single-index case, it provides a more fine-grained characterization of how the sample complexity of efficient recovery of the ground-truth direction depends on the signal-to-noise ratio.
This NSE-based characterization is simple and gives matching upper and lower bounds.

### Weaknesses

* It seems that the results are largely applications of (Barbier et al., 2019) and (Troiani et al., 2025). It might be worthwhile spending a paragraph or two explaining the technical contribution of the paper.
* The multi-index part of the paper is somewhat confusing:
  * First, the authors define a specialized solution to be any solution that is not the (uniform) average of the ground-truth weights. However, both the terminology "specialization" and the discussion in Section 6 (line 432) suggest that "specialization" means learning each individual ground-truth direction, which is not necessarily possible in this isotropic setting if we are going to rely on the second-order information (as in the definition of NSE), because of the rotation invariance of the second-order terms [1-2].
  * For the same reason (rotational invariance of the second-order terms), it is not surprising that when $\sigma$ is even, the weak recovery and specialization (in the sense of this paper) thresholds are the same.
  * When $\sigma$ is not even and IE is $1$, it is known that algorithms will converge to the average direction, and this is clear once we rewrite the loss as the sum of a sequence of tensor decomposition loss using Hermite analysis. (This might be a folklore result, but something similar can already be found in [3].) Therefore, it is also not surprising that in setting of Result 4.3, we cannot directly get specialized solutions. Probably a more proper question is, can we remove the influence of this average direction after learning it, so that the algorithm can then find specialized solutions? This is easy in the IE framework, but I am not sure if "removing" the first-order terms is possible under this NSE framework.


[1] Gérard Ben Arous, Cédric Gerbelot, Vanessa Piccolo. Langevin dynamics for high-dimensional optimization: the case of multi-spiked tensor PCA. 2024

[2] Yunwei Ren, Jason D. Lee. Learning Orthogonal Multi-Index Models: A Fine-Grained Information Exponent Analysis. 2024

[3] Yuanzhi Li, Tengyu Ma, Hongyang R. Zhang. Learning Over-Parametrized Two-Layer ReLU Neural Networks beyond NTK. 2020

---

> ### Author Rebuttal · Authors · 2026-03-30
>
> We thank the reviewer for their careful assessment and their feedback. Below, we address the raised questions.
>
> 1. **Technical contributions.** We appreciate the opportunity to clarify our precise technical contributions. Indeed, we build on the rigorous asymptotic characterization derived in Barbier et al. (2019), Aubin et al. (2018), and Troiani et al. (2025) for general single- and multi-index models. However, given these results, extracting an explicit, closed-form characterization of the dependence of the weak recovery transitions on the SNR is a highly non-trivial task. The critical thresholds derived from these frameworks are given by implicit equations involving $p$-dimensional integrals that must be evaluated and solved numerically for generic link functions.
>
>     The key technical contribution of our work lies in the analytical expansion of the critical threshold expression and the replica-symmetric potential, alongside the analysis of its minima in the high-noise limit. Through this analysis, we identify a fundamental property of link functions that governs the scaling of the weak recovery transitions, leading the observed statistical-to-computational gaps.
>
> 2. **On the committee machine's results.**
>
> - Indeed, following the established terminology in Aubin et al. (2018), our definition of an unspecialized solution exclusively refers to an estimator that correlates only with the uniform average of the ground-truth directions. Consequently, any solution that breaks this permutation symmetry is defined as specialized. We will clarify this terminology in the discussion in Section 6 to ensure a better consistency with our formal Definition 4.1.
> Hence, in this definition specialization is strictly weaker than identifiabiliy, which as the reviewer pointed out is not generally possible due to the presence of a rotational symmetry. Furthermore, while it might not appear surprising that specialization coincides with weak recovery for even committee machines under this definition, we stress that the primary claim of Theorem 4.2 is not the equivalence of these thresholds, but rather the fact that this specialization transition is governed by the NSE of the link function through a mechanism analogous to the one observed in high-noise single-index models.
>
> - First, we emphasize that our result does not only point out a separation between unspecialized and specialized learning phases in non-even committee machines, but rather shows that for models with NSE > 1, specialization cannot be efficiently achieved at the information-theoretic scaling $\alpha\asymp p$. This holds regardless of the first Hermite coefficient of the link function (i.e., whether IE = 1 or IE > 1).
>
>     For this reason, even removing the linear component in the Hermite expansion of the target, after perfectly learning the average direction, does not fundamentally alter the underlying phenomenology, as the AMP estimator would still correlate with the unspecialized solution at any sample complexity. In our framework, the existence of this unspecialized solution stems from the likelihood itself, as $\mathbb{E}_z[z P(y|z)]\propto{\bf 1}$ (see e.g., Troiani et al., (2025), and our Appendix D, lines 850-855), which takes into account the contributions from all odd terms in the Hermite expansion. Thus, in order to erase the unspecialized phase, one would need to preprocess the dataset such that $\mathbb{E}_z[z P(y|z)]={\bf 0}$, which is not straightforward to translate into practice.
>
>     Crucially, our mathematical derivation for non-even activations reflects exactly this logic: the model first reaches the unspecialized solution, and we then evaluate the local stability by expanding the replica-symmetric potential directly around this subspace. This represents the main technical difficulty in obtaining the precise size of the computational gap for the non-even case. Since this unspecialized subspace is structurally more complex than the uninformative fixed point ($q=0$) used in the even case, deriving a closed-form threshold is significantly harder; yet, the fundamental bottleneck induced by the NSE remains.

---

> > ### Author Rebuttal · Reviewer_cBtw · 2026-04-03
> >
> > I thank the authors for the clarification, especially the part on how removing the first-order terms or not doesn't affect analysis. I'll increase my score to 5.

---

### Official Review · Reviewer_8S4S · 2026-03-09

**Soundness:** 4
**Presentation:** 3
**Significance:** 3
**Originality:** 3
**Overall Recommendation:** 5
**Confidence:** 2

**Summary:**

This paper studies statistical-to-computational gaps in single-index models with Gaussian additive noise and separable multi-index models. The key contribution is the introduction of the noise sensitivity exponent (NSE), a quantity that serves as a unified principle across both settings. For single-index models, the NSE governs how the computational recovery threshold deteriorates as the signal-to-noise ratio decreases. For multi-index models, the NSE controls the sample complexity of weak recovery and specialization.

**Compliance With Llm Reviewing Policy:**

Affirmed.

**Final Justification:**

The rebuttal directly addresses my concerns about the comparisons between NSE and GE and how finite-dimensional effects may affect the results.

**Key Questions For Authors:**

1. Can you further explain how NSE improves the generative exponent in detail?
2. What are the challenges of quantifying finite-dimensional effects, and how would your results change when the dimension is finite?

I am open to increasing my score if the authors can successfully address the concerns raised above.

**Limitations:**

yes

**Strengths And Weaknesses:**

**Strengths**
This paper introduces noise sensitivity exponent (NSE), which controls the statistical-to-computational gaps in single- and multi-index models. This NSE quantity successfully reflects the order of the statistical-to-computational gaps in the proportional regimes.

**Weaknesses**
This paper focuses on a regime where the characterizations are asymptotic in the high-dimensional limit and does not quantify finite-dimensional effects.

---

> ### Author Rebuttal · Authors · 2026-03-30
>
> We thank the reviewer for their careful assessment and their feedback. Below, we address the raised questions.
> 1. Focusing for simplicity on the single-index model setting (eq. 4), the GE (Damian et al. (2024)) quantifies the polynomial scaling of the sample size $n$ in terms of the covariate dimension $d$ required for efficient weak recovery (specifically within the framework of statistical queries and low-degree polynomials). It establishes that for a given GE $k_\star$, efficient weak recovery requires $n \gtrsim d^{k_\star/2}$.
>     Within the class of even link functions, corresponding to $k_\star = 2$ (with the exception of fine-tuned examples), the NSE offers a strictly finer classification of models. By highlighting the role of the SNR, the NSE allows to prove the existence of large statistical-to-computational gaps otherwise invisible under the GE framework alone.
> 2.  The high-dimensional limit is the standard approach for the theoretical study of statistical-to-computational gaps, as it allows for exact analytical characterizations. Specifically, our entire mathematical framework relies on the concentration of correlations (overlap parameters) and "free-energy" functions around their expected values in high dimensions (see e.g., Barbier et al. (2019)). Quantifying exact finite-dimensional effects is a difficult mathematical challenge. Additionally, in a strictly finite-dimensional setting, it is not entirely clear how the exact theoretical results would change, and some of the questions become ill-posed. For instance, the sharp phase transitions we identify would smooth out into overlap scaling crossovers, and even a random estimator would exhibit a finite overlap of order $1/\sqrt{d}$ with the true signal direction, making the current definition of "weak recovery" ambiguous. Thus, it is highly non-trivial to conjecture that the computational hardness of correlating with the signal direction would follow analogous laws governed by the NSE.

---

> > ### Author Rebuttal · Reviewer_8S4S · 2026-04-03
> >
> > Thanks for the clarification. I will raise my score to 5.

---

### Official Review · Reviewer_zJAM · 2026-03-10

**Soundness:** 3
**Presentation:** 3
**Significance:** 2
**Originality:** 3
**Overall Recommendation:** 5
**Confidence:** 4

**Summary:**

This paper proposes a more fine-grained of the difficulty of learning a single-index model, and focuses on even activation functions which have generative exponent 2. They define the "noise sensitivity exponent" to be the smallest $\beta$ such that $\sigma^\beta$ has information exponent $2$. They prove that in an additive noise model, the computational weak detection threshold scales like $\frac{n}{d} \propto \mathtt{SNR}^{-\beta}$ and in a committee model, $\frac{n}{d} \propto \mathtt{width}^{\beta}$, and the paper discusses extensions to the specialization threshold.

**Compliance With Llm Reviewing Policy:**

Affirmed.

**Final Justification:**

I am satisfied by the authors' responses and maintain my positive assessment of the paper.

**Key Questions For Authors:**

- Based on the heuristic explanation under equation 11, I would expect this to naturally generalize to the following claim: Let $y = \sum_i \sqrt{\lambda_i} \cdot \sigma(w_i \cdot x) + \xi$ with the $\{\lambda_i\}$ sorted. Then the "signal" for the first direction is $\sqrt{\lambda_1}$ and the "noise" is $\sqrt{1 + \sum_{i>1} \lambda_i}$, which suggests that you would need $\alpha \gtrsim \left(\frac{1 + \sum_{i > 1} \lambda_i}{\lambda_i}\right)^\beta$. Is this the right way to unify the two models?
- Is there a way to generalize this intuition to general multi-index models? For example, if $W^\star$ is sequentially learned by AMP and $w_1^\star$ is the first direction, then could recovering $w_1^\star$ be viewed as a noisy single-index model in which everything orthogonal to $w_1^\star$ is noise?

**Limitations:**

yes

**Strengths And Weaknesses:**

Strengths:
- The derivations are very clean and leverage existing machinery for the weak recovery thresholds for AMP
- The paper's claim that $\beta$ is a less brittle/more fine grained hardness parameter compared to the generative exponent is reasonable
- The paper studies both noisy single-index models and noiseless committee machines and shows they are controlled by the same exponent

Weaknesses:
- The analysis is specialized to isotropic Gaussian inputs (although this is fairly standard in the field)
- There is no discussion of how this exponent might extend to arbitrary multi-index models beyond the committee machine

---

> ### Author Rebuttal · Authors · 2026-03-30
>
> We thank the reviewer for their careful assessment and appreciation of our work.
>
> - As pointed out, the assumption of isotropic Gaussian inputs is a standard setting in the theoretical literature. This choice deeply connects our model to existing work and ensures its analytically tractability. We have explicitly acknowledged this assumption and its context within the field in our discussion (see Section 6, from line 390).
>
> - As suggested by the reviewer, a natural extension of our results concerns the proposed additive multi-index models, analogous to the targets investigated in [1] in the context of one-pass SGD on shallow neural networks. We plan to include a short discussion on this model in the revised version. For such additive targets $y = \sum_{i=1}^{p}\sqrt{\lambda_{i}} \sigma(\mathbf{x}^{\top} \cdot \mathbf w_{i}^{\star}) + \xi,$ it can be shown that the weak recovery of the $i$-th direction is achieved at sample complexities $\alpha \gtrsim \lambda_{i}^{-\beta_{\star}},$ with unlearned directions acting as effective noise, and the NSE $(\beta_{\star})$ still governing the computational bottleneck and inducing a statistical-to-computational gap.
> - Extending this analysis to a fully general class of multi-index models, however, would require a rigorous analysis of multivariate state evolution equations (Aubin et al. (2018); Troiani et al. (2025)). This is a challenging task due to the emergence of non-trivial correlations between the sequentially learned directions. Nevertheless, our current findings strongly suggest that the NSE might play a fundamental  role in the efficient weak recovery of low-SNR features even in these more generic contexts.
>
>
> [1] Ren et al. "Emergence and scaling laws in SGD learning of shallow neural networks", 2025

---

> > ### Author Rebuttal · Reviewer_zJAM · 2026-04-04
> >
> > Thank you for answering my questions. I would like to maintain my positive score.

---

### Official Review · Reviewer_5ezq · 2026-03-12

**Soundness:** 3
**Presentation:** 3
**Significance:** 2
**Originality:** 4
**Overall Recommendation:** 5
**Confidence:** 3

**Summary:**

This paper studies statistical–computational gaps in high-dimensional regression problems and introduces a structural quantity, the noise sensitivity exponent (NSE), which characterizes the computational difficulty of learning in single-index and multi-index models. The main contribution is the identification of NSE as a unifying parameter governing when polynomial-time algorithms can succeed and when a statistical–computational gap emerges.

The paper establishes that the sample complexity required by polynomial-time algorithms scales as O (d^NSE), up to logarithmic factors, and demonstrates how this exponent controls the gap between statistical feasibility and computational tractability. Importantly, the framework applies to both single-index and multi-index models.

**Compliance With Llm Reviewing Policy:**

Affirmed.

**Final Justification:**

This is a good paper. I maintain my score.

**Key Questions For Authors:**

See weakness: can you comment on the potential impact/applicability of these results?

**Limitations:**

yes

**Strengths And Weaknesses:**

Strengths:

1.Conceptual contribution.
The introduction of the noise sensitivity exponent as a structural parameter governing computational hardness is a clean and compelling idea. It provides a unifying lens through which statistical–computational gaps in regression models can be understood.

2. Strong theoretical results.
The paper derives bounds linking NSE to the sample complexity achievable by polynomial-time algorithms, offering a principled characterization of when efficient learning is possible.

3. Clear positioning in the literature.
The paper connects well with prior work on statistical–computational tradeoffs, low-degree methods, and noise stability, and clearly explains how its contribution fits into this line of research.

4. Good exposition.
For a technically involved theory paper, the presentation is quite clear. The problem setting and the key quantities are introduced early, and the structure of the results is easy to follow.

Weaknesses:

1.Limited impact/significance
A potential limitation is that the significance of the results may remain somewhat specialized to the theoretical study of statistical–computational gaps in high-dimensional regression models. Broader implications for practical machine learning settings or more complex model classes are not yet fully clear. However, I think the level of significance remains in line with what is expected for a theory paper.

---

> ### Author Rebuttal · Authors · 2026-03-30
>
> We thank the reviewer for their appreciation of our work and for acknowledging that our contributions align with the expectations for a theoretical paper.
>
> While bridging the gap to real-world problems remains an open challenge, our framework can in principle be extended to more structured models. For instance, we refer to our response to Reviewer zJAM regarding additive multi-index models, for which the NSE governs the scaling of the weak recovery threshold for each individual direction, with the remaining components acting as effective noise, altering the phenomenology highlighted by [1,2] in the context of quadratic networks. We will emphasize these broader implications and extensions in the revised discussion.
>
> [1] Ben Arous et al. "Learning quadratic neural networks in high dimensions: SGD dynamics and scaling laws", 2025
>
> [2] Defilippis et al. "Scaling Laws and Spectra of Shallow Neural Networks in the Feature Learning Regime", 2025

---

> > ### Author Rebuttal · Reviewer_5ezq · 2026-04-02
> >
> > I am satisfied with the response of the authors and I will maintain my score.

---

### Decision · Program_Chairs · 2026-04-30

**Decision:**

Accept (spotlight)

**Comment:**

The paper studies statistical-computational gaps in learning single index and committee models (a special case of multi-index models). In particular, they study learning activation functions which have generative exponent 2 and learning over the gaussian distribution. Their main contribution introduce a new quantity, a noise sensitivity exponent that depends only on the activation function, and show that this characterizes the weak detection threshold for computationally efficient algorithms.

The reviewers and I unanimously found the paper well-motivated and well-written. I especially appreciated that the authors did a good job explaining the landscape of work on feature learning and situating their work within it. The technical results are strong with clean proofs, and the proposed NSE could turn out to be an interesting quantity for future work in this area, beyond the generative exponent. Therefore, I recommend accept.